# Conditional control of fluorescent protein degradation by an auxin-dependent nanobody

Katrin Daniel[1], Jaroslav Icha [2,3], Cindy Horenburg[1], Doris Müller[1], Caren Norden [2] & Jörg Mansfeld [1]

The conditional and reversible depletion of proteins by auxin-mediated degradation is a powerful tool to investigate protein functions in cells and whole organisms. However, its wider applications require fusing the auxin-inducible degron (AID) to individual target proteins. Thus, establishing the auxin system for multiple proteins can be challenging. Another approach for directed protein degradation are anti-GFP nanobodies, which can be applied to GFP stock collections that are readily available in different experimental models. Here, we combine the advantages of auxin and nanobody-based degradation technologies creating an AID-nanobody to degrade GFP-tagged proteins at different cellular structures in a conditional and reversible manner in human cells. We demonstrate efficient and reversible inactivation of the anaphase promoting complex/cyclosome (APC/C) and thus provide new means to study the functions of this essential ubiquitin E3 ligase. Further, we establish auxin degradation in a vertebrate model organism by employing AID-nanobodies in zebrafish.

[1] Cell Cycle, Biotechnology Center, Technische Universität Dresden, Tatzberg 47-49, 01307 Dresden, Germany. [2] Max Planck Institute of Molecular Cell Biology and Genetics, Pfotenhauerstraße 108, 01307 Dresden, Germany. [3] Present address: Turku Centre for Biotechnology, University of Turku, Tykistokatu 6, 20520 Turku, Finland. These authors contributed equally: Katrin Daniel, Jaroslav Icha. Correspondence and requests for materials should be addressed to C.N. (email: norden@mpi-cbg.de) or to J.M. (email: joerg.mansfeld@tu-dresden.de)

Conditional loss of function studies are fundamental to reveal specific protein functions in complex biological systems. The rapid degradation of proteins fused to an auxin-inducible degron (AID) enables the generation of conditional knockdowns at the protein level[1–4] and thus belongs to the few approaches that enable determination of acute phenotypes in a reversible manner. Degradation requires the ectopic expression of the plant F-Box protein TIR1, which recruits proteins tagged with AID in an auxin-dependent manner to the SKP1-CUL1-F-Box (SCF) ubiquitin E3 ligases resulting in their ubiquitylation and proteasomal degradation. A caveat with this approach is however the need for genetic engineering as the AID needs to be site-specifically inserted into the alleles of each targeted protein. Further, it has been reported that fusion with the AID degron can destabilize the tagged protein[4]. So far, the auxin system has been established in a limited number of case studies including yeast[4], nematodes[5], flies[1], and human cell lines[3,6,7]. However, its feasibility in a vertebrate model organism remains to be shown and large-scale application of the AID system in animal remains challenging despite the advent of CRISPR/Cas9.

deGradFP is an alternative approach to target proteins for degradation[8] and takes advantage of genetically encoded nanobodies that can recognize GFP-tagged proteins in living cells[9]. deGradFP employs a fusion of the anti GFP-nanobody vhhGFP4[10] to the F-box domain of the F-box protein Slimb enabling direct and effective GFP-fusion protein removal in a SCF and proteasome-dependent manner in flies and human cell culture[8]. As the efficiency of deGradFP degradation has been shown to differ between model organisms deGradFP-like approaches that employ distinct degradation domains have been developed in nematodes[11] and zebrafish[12]. One advantage compared to the AID system are stock collections of endogenous proteins tagged with GFP or GFP-like proteins (e.g., YFP, Venus, and Citrine), which are recognized by anti-GFP nanobodies. Such collections are readily available in model systems such as flies and zebrafish[13–15] and endogenous GFP-fusions are also increasingly used in human cell lines (this study[16–19]). Hence, degradation technologies targeting GFP have the potential to become a widespread application in a multitude of experimental systems, especially in animal model organisms, due to the possibility to obtain homozygous GFP-trap alleles by breeding.

Compared to the AID system however, deGradFP and related nanobody-mediated degradation systems suffer from two key disadvantages. First, the induction of degradation is coupled to the de novo expression of the nanobody-F-box fusion and therefore only provides a rough temporal control. Second, degradation is not reversible as long as the nanobody-degron fusion protein is present, thus precluding experiments that depend on the transient inactivation of the target protein.

We reasoned that merging the two elements that provided reversibility of AID and specificity of nanobody-dependent degradation would mitigate disadvantages of both technologies and provide a potent alternative degradation tool to address biological questions from the cellular to the organismal level. We show that expression of a customized AID-nanobody fusion in combination with TIR1 provides a powerful strategy to reversibly deplete GFP-tagged proteins localized to distinct cellular structures by ubiquitin-mediated proteolysis in an auxin-dependent manner. Comparing AID-nanobody-mediated degradation with established auxin and deGradFP technologies, we find that successful application and degradation efficiency of each system is context-dependent and differs for individual target proteins. By targeting endogenous ANAPC4, an essential subunit of the anaphase promoting complex/cyclosome (APC/C), we provide an example for which only the mAID-nanobody technology enables a reversible functional inactivation of this crucial cell cycle enzyme. Finally, we show that the auxin system can be applied to a vertebrate model organism by demonstrating effective degradation of GFP-tagged proteins by mAID-nanobodies in zebrafish.

## Results

**Development of a lysine-less mAID-nanobody.** Protein degradation of GFP-fusion proteins by the auxin-inducible nanobody system (Fig. 1) requires ectopic expression of the plant F-box protein TIR1 and an anti-GFP nanobody fused to the AID degron. To ensure efficient protein degradation in both, the nucleus and the cytoplasm, we first created HeLa and retina pigment epithelial cell lines (hTERT RPE-1) stably expressing TIR1 either tagged with a nuclear localization sequence (NLS) or a nuclear export sequence (NES) (Supplementary Figure 1a). To reduce the size of the AID-nanobody fusion, we first explored the potential of single (64 aa) and triple (192 aa) minimized AID (mAID) sequences reported to have a similar or superior efficiency compared to the full length AID[20] (Supplementary Figure 1b). While full length AID and single mAID did not affect the localization of Venus, the Venus-triple mAID fusion induced aggregate formation in HeLa cells (Supplementary Figure 1c). Consequently, we fused a single mAID labeled with a hemagglutinin (HA) epitope for detection onto the N-terminus of the anti-GFP nanobody vhhGFP4[8,10] creating mAID-vhhGFP4 (Supplementary Figure 1d). To assess the stability of the mAID-nanobody fusion in response to auxin treatment with indole-3-acetic acid (IAA) or the synthetic auxin analog 1-naphtaleneacetic acid (NAA, thereafter generally referred to as auxin), we inserted tetracycline-inducible mAID-vhhGFP into the TIR1-expressing HeLa cell line. Western blot analysis revealed substantial depletion of mAID-vhhGFP4 in the presence of auxin (Supplementary Figure 1e), indicating that SCF[TIR1] ubiquitylated mAID-vhhGFP4 on one or several of the lysine residues present in mAID and/or vhhGFP4. We reasoned that replacing all lysine

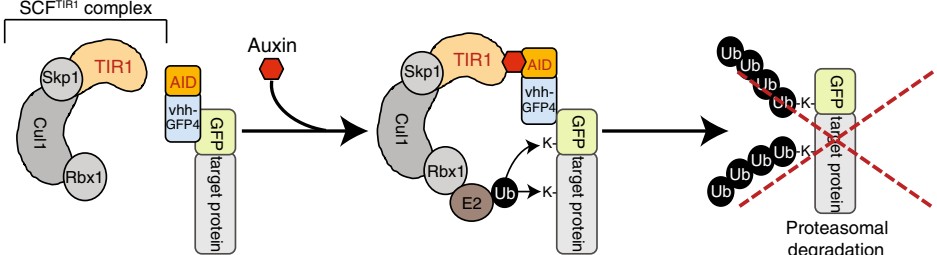

**Fig. 1** Auxin-dependent degradation of GFP-tagged proteins by an anti-GFP nanobody. Schematic illustration of AID-nanobody degradation. GFP-tagged proteins are recognized by an AID-nanobody fusion. Addition of auxin triggers recruitment of the SCF[TIR1] ubiquitin E3 ligase. Ubiquitylation of the GFP fusion protein occurs at lysine residues present in GFP and/or the target protein leading to proteasomal degradation. AID, auxin-inducible degron; Cul1, Cullin-1; Rbx1, RING-box protein 1; Skp1, S-phase kinase-associated protein 1; TIR1, transport inhibitor response 1; vhh-GFP4, anti-GFP nanobody

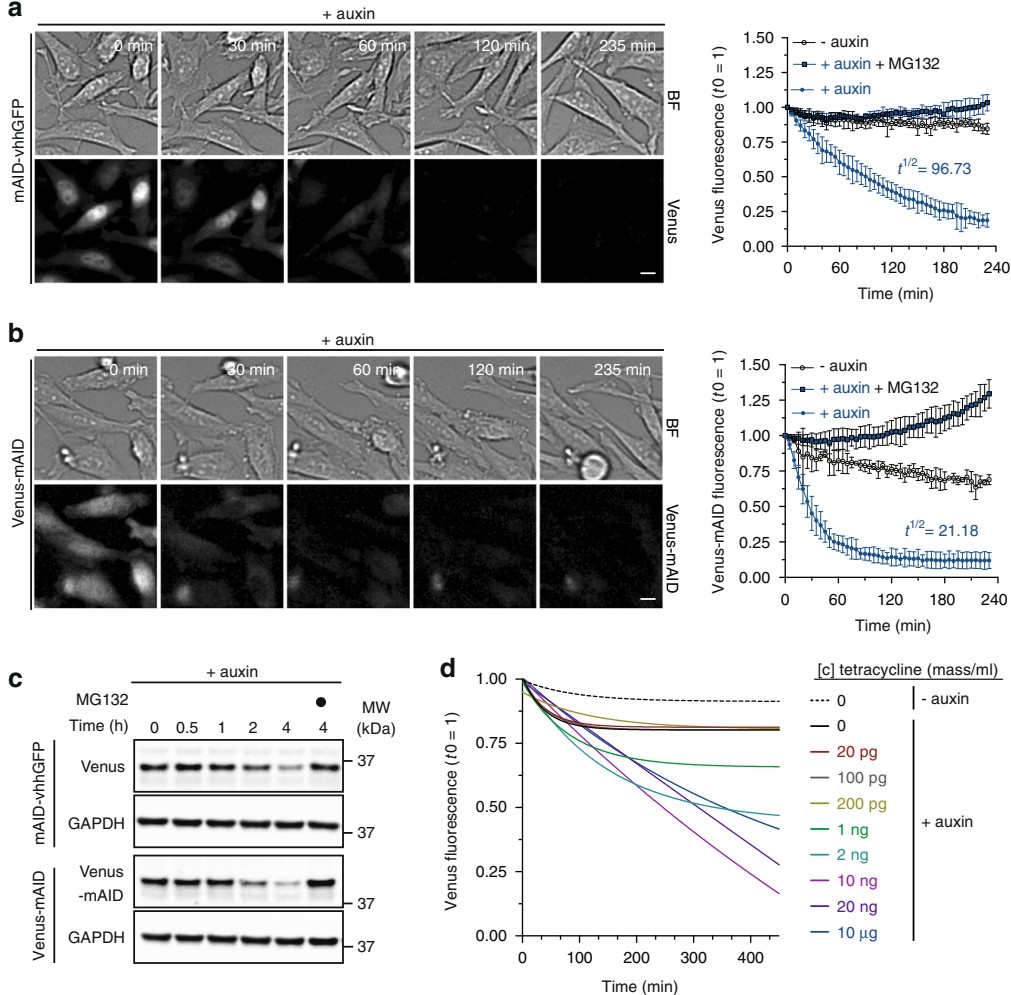

**Fig. 2** Kinetics of Venus degradation by mAID-nanobodies. **a** Time-lapse images and degradation kinetics of Venus targeted by the mAID-nanobody or **b** of Venus-AID after the addition of 0.5 mM auxin. Scale bar: 10 μm. Quantifications show the mean ± s.d. ($n = 5$, -auxin, +auxin +MG132; $n = 10$, +auxin). **c** Western blot analyses of cells treated as in **a** and **b** for the indicated time with 0.5 mM auxin or auxin together with 10 μM proteasome inhibitor MG132. **d** Titration of tetracycline-inducible expression of mAID-nanobodies and the effect on the degradation kinetics of Venus in the presence of 0.5 mM auxin. Note, the 100 pg curve is masked by the 20 and 0 pg curves. Quantifications show non-linear fit (one-phase decay) of ($n = 6$, 0 tetracycline ± auxin; $n = 3$, remaining concentrations). Note, images of –auxin, +auxin and 10 μM MG132-treated cells, degradation of Venus by deGradFP, and Western blot detection of mAID-nanobody expression during the titration are shown in Supplementary Figure 2

residues present in mAID-vhhGFP4 with structurally related arginine residues that cannot be ubiquitylated could provide a strategy to stabilize the mAID-vhhGFP4 and to direct SCF$^{TIR1}$ activity exclusively towards GFP-fusion proteins. To examine if lysine to arginine mutagenesis has a functional impact on the ability of TIR1 to recognize mAID, we replaced all lysines present in mAID but not in vhhGFP4 (mAID$^{K\rightarrow R}$-vhhGFP4) (Supplementary Figure 1d). The lysine-less mAID still supported degradation of the mAID$^{K\rightarrow R}$-vhhGFP4, presumably due to ubiquitylation of the remaining lysines present in vhhGFP4 (Supplementary Figure 1e). This finding suggested that lysine to arginine mutagenesis did not impair the recognition of lysine-less mAID by TIR1. Mutagenesis of all lysines in vhhGFP4 further stabilized mAID$^{K\rightarrow R}$ vhhGFP4$^{K\rightarrow R}$ (hereafter generally referred to as mAID-nanobody or mAID-vhhGFP) (Supplementary Figure 1e).

To assess the functionality of the complete lysine-less mAID-nanobody, we monitored its ability to degrade the GFP-like protein Venus. Adding auxin to a stable cell line expressing Venus and tetracycline-inducible mAID-nanobody resulted in Venus degradation in an auxin- and proteasome-dependent

manner (Fig. 2a, Supplementary Figure 2a). Degradation of Venus indicated that lysine mutagenesis in vhhGFP4 is not critical for nanobody binding to GFP or GFP-like proteins.

Thus, we conclude that due to its ability to solely target the fluorescent protein, the mAID-nanobody is in principle able to degrade any GFP-tagged protein independently of the availability of additional lysine residues present within the target protein.

**Kinetics of Venus degradation by mAID-nanobodies**. We observed a depletion of Venus by the mAID-nanobody with a half-life of ~97 min (Fig. 2a) and thus tested how this relates to degradation kinetics of a direct AID-Venus fusion protein. We fused wildtype mAID to the C terminus of Venus and quantified degradation efficiency by life cell imaging. Compared to degradation by the mAID-nanobody the direct fusion of mAID to Venus (Venus-mAID) resulted in ~4-fold faster degradation kinetics ($t^{1/2} = 21.18$ min) (Fig. 2a, b, Supplementary Figure 2a). This result was confirmed by Western blot analysis (Fig. 2c). This difference might reflect the reduced complexity of a 2-component degradation system compared to our 3-component approach. Alternatively, it might be a consequence of the additional lysine

residues present in wildtype mAID providing more potential ubiquitylation sites.

Next, we investigated how different expression levels of the mAID-nanobody affect degradation kinetics. Therefore, we titrated mAID-nanobody expression over a seven-fold range in a using increasing tetracycline concentrations (Fig. 2d, Supplementary Figure 2b) Time-lapse imaging after auxin addition revealed that differences in the levels of mAID-nanobody had a minor impact on the efficiency of Venus degradation (Fig. 2d) once the levels of mAID-nanobody and the levels of Venus were in the same range (Fig. 2d, Supplementary Figure 2b). Levels of mAID-nanobody that were below Venus levels resulted in partial degradation. We observed that at its highest expression the mAID-nanobody slightly slowed down Venus degradation kinetics (Fig. 2d, Supplementary Figure 2b), possibly reflecting that an excess of free mAID-nanobody competes with target-bound nanobodies for binding to SCF$^{TIR1}$ (Fig. 2d, Supplementary Figure 2b).

Taken together, we conclude that the mAID-nanobody robustly targets Venus for degradation when enough nanobody is expressed, albeit at a slower rate than achieved by a direct Venus-mAID fusion. These results show that our approach is a powerful tool for the inducible degradation of GFP and GFP-like proteins.

**mAID-nanobodies function in different cellular compartments**. So far, we have shown that the mAID-nanobody enables degradation of the Venus protein, which diffuses freely in the nucleus and cytoplasm. To determine whether mAID-nanobodies are able to also target GFP-fusion proteins present at distinct subcellular locations and to assess if auxin removal allows protein recovery including re-established protein localization, we created three TIR1 cell lines expressing different fluorescent fusion proteins: (i) Venus-tagged Lamin A (Venus-LMNA), which integrates into the nuclear lamina, a complex filamentous meshwork beneath inner nuclear membrane (Fig. 3a); (ii) Venus-tagged Cyclin D1 (CCND1-Venus), which localizes to both, the cytoplasm and nucleus (Fig. 3b), and (iii) GFP-tagged Inner centromere protein INCENP (INCENP-GFP), which localizes to distinct DNA-associated structures (Fig. 3c). The fact that INCENP-GFP is expressed from the endogenous promoter in context of a bacterial artificial chromosome (BAC)[21] further allowed us to evaluate degradation and recovery of an endogenously regulated protein in comparison to CMV promotor-driven Venus-LMNA and CCDN1-Venus.

To enable degradation, mAID-nanobody expression was either induced by tetracycline addition (Venus-LMNA and CCDN1-Venus) or by transient transfection of a construct expressing the mAID-nanobody and nuclear-localized mCherry as a transfection control from the same mRNA. In all three cases, time-lapse imaging revealed extensive and efficient degradation of the target protein after auxin addition as well as gradual recovery of the proteins following auxin removal (Fig. 3a–c). Notably, CMV-driven Venus-LMNA and CCDN1-Venus recovered slower than INCENP that was expressed under the control of endogenous promotor. This likely reflects differences in mRNA levels, translation efficiency and/or folding properties in between these proteins.

Quantification of substrate depletion showed that kinetics of auxin-dependent degradation varied among the analyzed target proteins: while CCDN1-Venus was degraded with a half-life of ~84 min, 50% depletion of Venus-LMNA required ~59 min and only ~21 min for INCENP-GFP (Fig. 3d). Of note, in all cases the fusion protein was degraded more efficiently than the Venus tag itself (Fig. 2a), suggesting that additional ubiquitylation sites

present in the protein fused to GFP/Venus positively affected degradation kinetics. During degradation of INCENP-GFP we observed a population of cells with incomplete degradation and slower kinetics ($t^{1/2} = 28.87$ min). These might reflect cells in which the transiently expressed nanobody became limiting during the experiment (Figs. 2, 3c).

We conclude that mAID-nanobody-mediated degradation can be achieved for GFP-fusion proteins in different cellular compartments. We observed rapid but individually varying degradation kinetics up to the rate observed for Venus-mAID (Fig. 2b) or direct substrate-AID fusions in mammalian cells[3]. Importantly, the recovery of all substrates after auxin removal confirmed the reversibility of protein depletion in the presence of mAID-nanobodies.

**mAID-nanobodies rapidly and reversibly inactivate the APC/C**. A major potential application for mAID-nanobodies is the conditional and reversible inactivation of endogenous GFP-tagged proteins to reveal acute phenotypes of otherwise essential proteins. In addition, the ability to correlate the degree of depletion and the emerging phenotype in individual cells by microscopy can facilitate the analysis of cause-effect relationships. As a proof of principle for the inactivation of an endogenous protein by the mAID-nanobody, we chose ANAPC4 as a target. The following rationales drove this decision: (i) ANAPC4 is a subunit of the APC/C, which allows assaying recognition by mAID-nanobody even in the context of a large and compact 15-subunit protein complex of 1.5 MDa. (ii) APC/C-dependent destruction of mitotic cyclins A2 (CCNA2) and B1 (CCNB1) is essential for mitotic progression and consequently its efficient inactivation yields an easily detectable mitotic metaphase arrest once cells enter mitosis.

To enable ANAPC4 degradation by mAID-nanobodies, we inserted a 3x Flag-streptaviding binding peptide (SBP)-Venus tag into all alleles of the *ANAPC4* gene in a HeLa cell line stably expressing TIR1 and tetracycline-inducible mAID-nanobody by CRISPR/Cas9-assisted homologous recombination (Supplementary Figure 3a). Neither, expression of Flag-SBP-Venus-ANAPC4 or TIR1 alone nor of Flag-SBP-Venus-ANAPC4, TIR1 and mAID-nanobody together caused a growth defect compared to the parent cell line (Supplementary Figure 3b). Furthermore, the intracellular localization of endogenous Flag-SBP-Venus-ANAPC4 (from here, Venus-ANAPC4) did not differ from the localization of untagged endogenous ANAPC4 (Supplementary Figure 3c) and Venus-ANAPC4 incorporated equally well as the untagged ANAPC4 into the APC/C (Supplementary Figure 3d). Together, this showed that the fusion protein is functional.

Quantitative spatio-temporal analysis of Venus-ANAPC4 levels in single cells monitored over 3 h after auxin supply showed efficient depletion of the protein (Fig. 4a and Supplementary Movie 1). With a half-life of ~16 min, degradation in the nucleus was slightly more efficient than in the cytoplasm (28 min) (Fig. 4b). Quantitative Western blot analysis confirmed the degradation kinetics established by live cell imaging (Supplementary Figure 3e, f) and suggested that degradation indeed removed the entire protein as judged by anti-ANAPC4 or anti-GFP detection (Supplementary Figure 3g). Depleting Venus-ANAPC4 for 3 h partially affected the protein levels of ANAPC2, ANAPC10 and ANAPC11, whereas the protein levels of all other tested APC/C subunits remained largely unchanged (Supplementary Figure 3h). Judging APC/C overall integrity after Venus-ANAPC4 depletion by immunoprecipitating the unaffected ANAPC3 subunit indicated that removal of ANAPC4 caused a partial disassembly of the holoenzyme (Supplementary Figure 3i). The observed decrease in protein levels of some

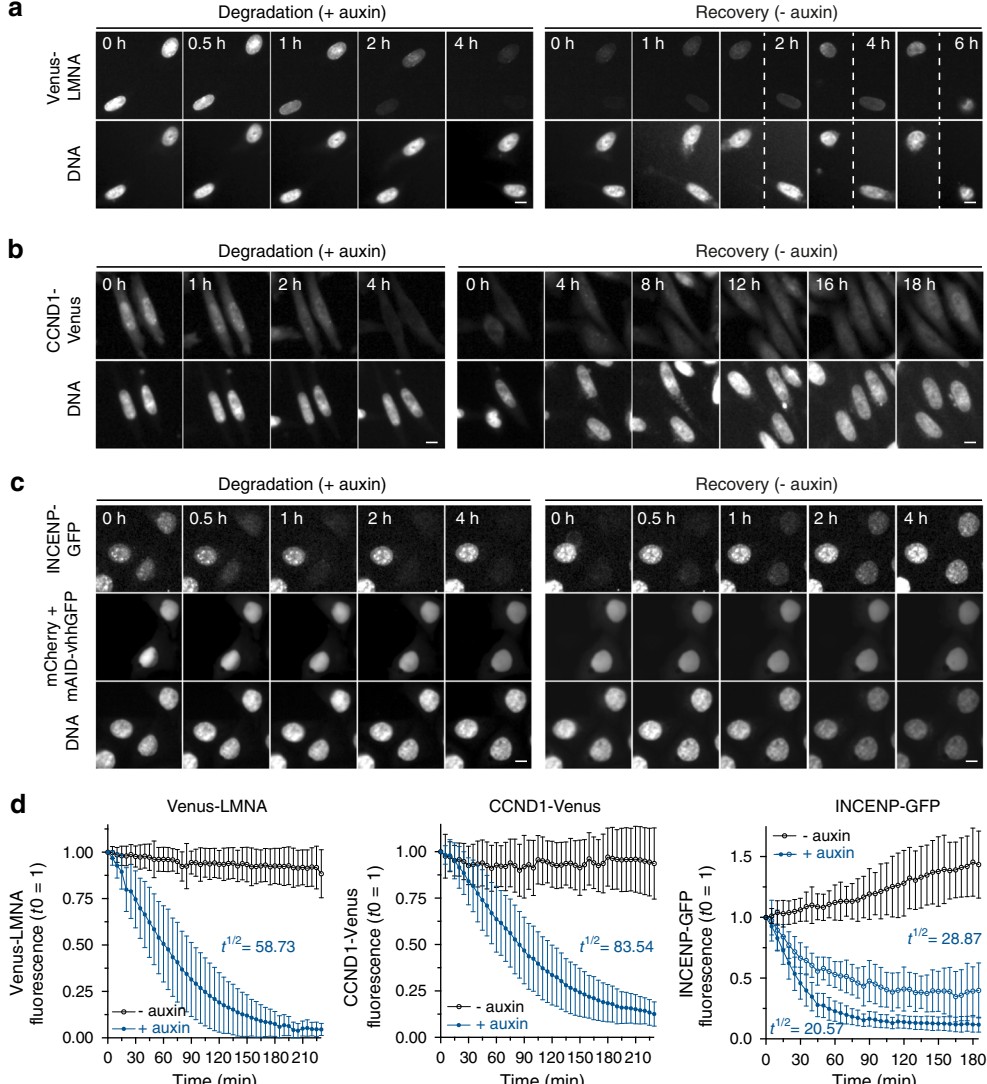

**Fig. 3** Conditional and reversible depletion of GFP-fusion proteins localized to distinct cellular compartments and structures by mAID-nanobodies. **a** Time-lapse images showing the degradation and recovery of (**a**) nuclear lamina-associated Venus-LMNA and **b** nuclear and cytoplasmic CCND1-Venus in Hela cell lines expressing tetracycline-inducible mAID-nanobody or **c** the centromere-associated Inner centromere protein (INCENP-GFP) after transient transfection of mAID-nanobody and nuclear mCherry expressed from the same mRNA (mCherry + mAID-VHH, see Methods). **a–c** Degradation and recovery were induced by the addition or removal of 0.5 mM auxin for the indicated time. Scale bar: 10 μm. **d** Single cell degradation kinetics of Venus-LMNA, CCND1-Venus and INCENP-GFP from the experiments shown in **a–c**. Open blue circles indicate degradation INCENP-GFP cells that reach a plateau at ~50% of the initial fluorescence. Quantifications show the mean ± s.d. Venus-LMNA and CCND1-Venus ($n = 40$, –auxin; $n = 40$, +auxin). INCENP-GFP ($n = 14$, –auxin; $n = 29$, +auxin (plateau); $n = 40$, +auxin)

APC/C subunits might reflect a reduced protein stability, which can be observed after disrupting the integrity of large molecular assemblies[22,23]. However, we cannot exclude that the SCF[TIR1] ubiquitylated in addition to ANAPC4 also neighboring APC/C subunits.

To investigate whether the degree of ANAPC4 depletion was sufficient to interfere with APC/C function, we monitored the accumulation of mitotic cells in asynchronously growing cells for 24 h. In agreement with a strongly prolonged duration of mitosis (Supplementary Figure 3j), we observed an increasing number of cells that were arrested at metaphase (Fig. 4c). In addition, we noted elevated protein levels of mitotic cyclins CCNB1 and CCNA2 indicative of a mitotic arrest (Fig. 4d). Thus, the degradation of Venus-ANAPC4 by the mAID-nanobody efficiently abrogated the main function of the APC/C. Finally, we assessed if removal of auxin allowed Venus-ANAPC4 recovery

and functional reconstitution of APC/C. Indeed, already 8 h after auxin removal Venus-ANAPC4 levels recovered sufficiently to allow progression through mitosis (Fig. 4e) and pre-treatment Venus-ANAPC4 levels were reached after prolonged recovery (Fig. 4f, g).

We conclude that functional recovery of proteins after mAID-nanobody depletion is possible also for components of multi subunit protein complexes that partially disintegrate upon target protein depletion.

**Comparing mAID-nanobody with AID and deGradFP systems.** A direct fusion of mAID to Venus enabled faster degradation kinetics than using the mAID-nanobody in context of Venus alone (Fig. 2). However, monitoring the degradation of fluorescent fusion proteins revealed that mAID-nanobodies can achieve similar degradation rates (Figs. 3 and 4). To compare the

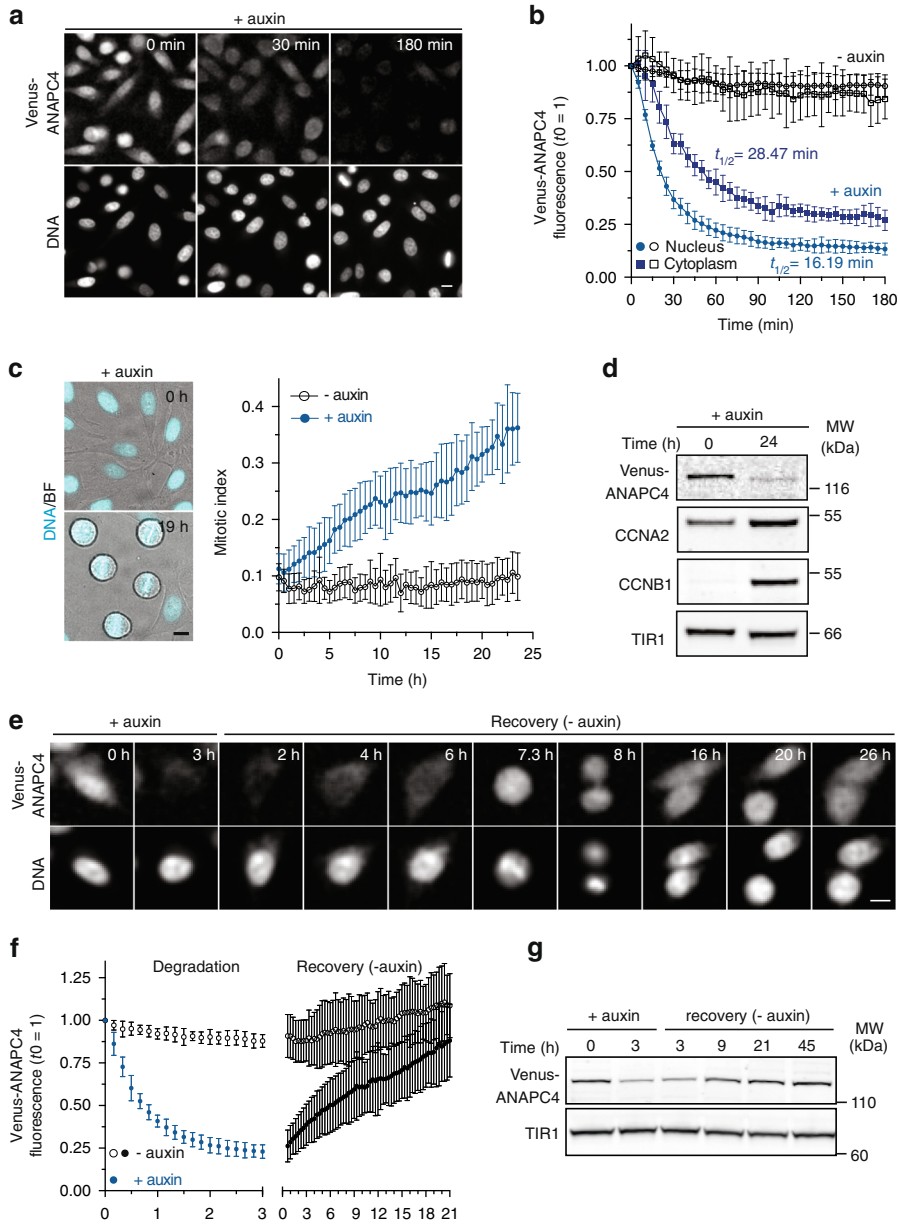

**Fig. 4** mAID-nanobodies allow the rapid but reversible inactivation of the APC/C. **a** Degradation of endogenously tagged Venus-ANAPC4 after the addition of 0.5 mM auxin. Scale bar: 10 μm. ANAPC4, APC/C subunit 4. See also Supplementary Figure 3 for characterization of the ANAPC4 knock-in. **b** Quantification of Venus-ANAPC4 fluorescence from time-lapse data in **a** showing the kinetics of Venus-ANAPC4 degradation in the nucleus and cytoplasm represented as the mean ± s.d. ($n = 6$, all conditions). **c** Time-lapse images and quantification of Venus-ANAPC4 cells arresting in mitosis in the presence of 0.5 mM auxin. Data represent the mean ± s.d. ($n = 12$, −auxin and +auxin). Scale bar: 10 μm. **d** Western blot analysis of cells treated with 0.5 mM auxin for the indicated time. Accumulation of cyclins CCNA2 and CCNB1 are indicative of a mitotic arrest. TIR1 serves as a loading control. **e** Time-lapse images of Venus-ANAPC4 cells treated for 3 h with 0.5 mM auxin followed by auxin removal and 26 h of recovery. Note, Venus-ANAPC4 recovery after 7.3 h is sufficient to allow mitotic progression. Scale bar: 10 μm. **f** Quantification of Venus-ANAPC4 recovery from cells treated as in **e** represented as mean ± s.d. ($n = 18$, −auxin; $n = 39$, +auxin). Scale bar: 10 μm. **g** Western blot analysis of cells treated as in **e**, TIR1 serves as a loading control

degradation potential of mAID-nanobodies to a direct mAID-fusion in a biologically relevant context we assessed Venus-ANAPC4 depletion and functional inactivation of the APC/C. We generated an endogenous mAID-Venus-ANAPC4 fusion in a cell line stably expressing TIR1 by CRISPR/Cas9-assisted homologous recombination (Supplementary Figure 4a). Western blot analysis of total cell lysates derived from two clones confirmed the expression of mAID-Venus-ANAPC4 (Supplementary Figure 4b) and immunoprecipitation with anti-ANAPC3 antibodies indicated that the fusion protein incorporated similarly well as

untagged ANAPC4 into the APC/C (Supplementary Figure 4c). Of note, we observed mild differences in the expression level and intracellular localization of mAID-Venus-ANAPC4 compared to Venus-ANAPC4 and untagged ANAPC4. mAID-Venus-ANAPC4 levels were slightly higher and the protein localized more prominently to the cytoplasm (Supplementary Figures 4d and 3c). It is unlikely that this is due to the molecular size of the epitope tag on ANAPC4 because the mAID-Venus tag contains only four additional amino acids compared to the Flag-SBP-Venus tag. Indeed, Flag-SBP-Venus-ANAPC4 and

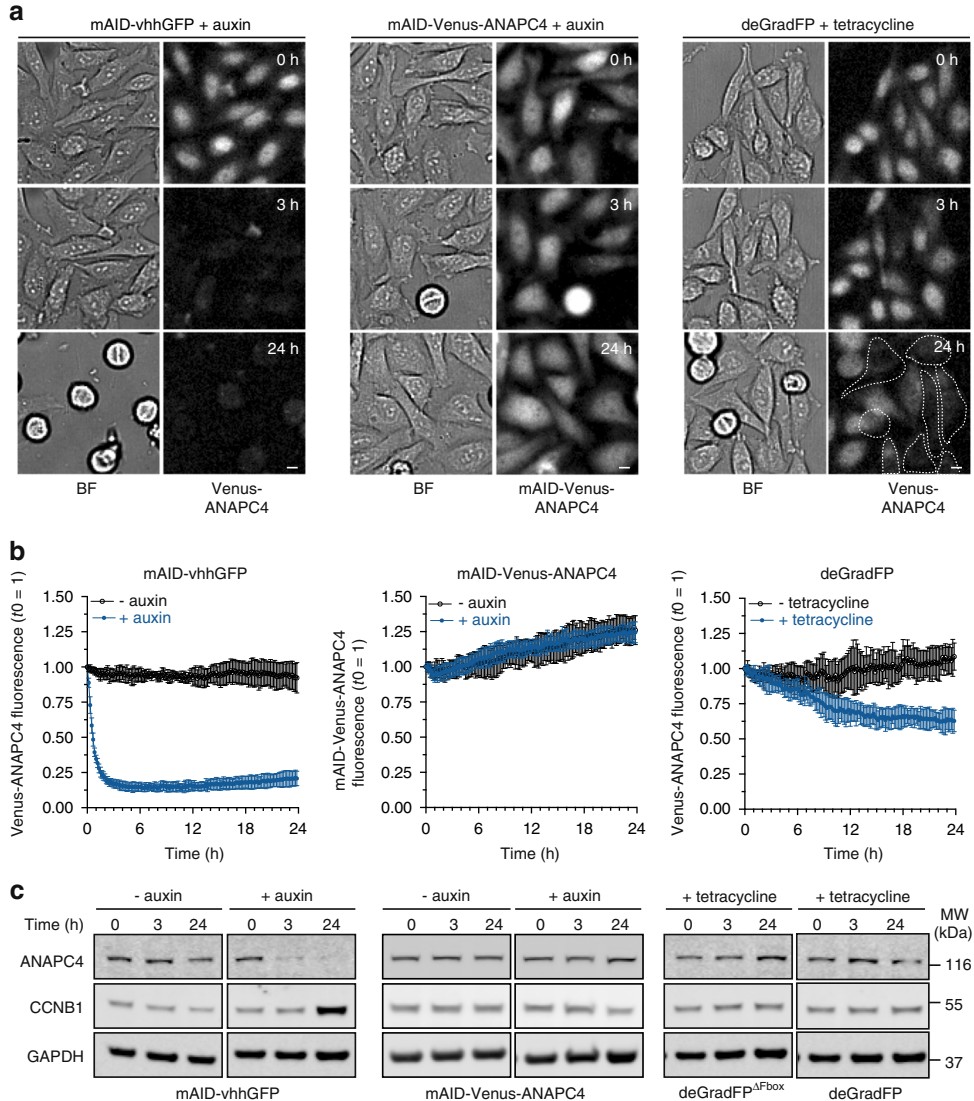

**Fig. 5** Degradation of endogenous ANAPC4 by mAID-nanobodies in comparison with AID and deGradFP systems. **a** Time-lapse images showing the degradation of endogenously tagged Venus-ANAPC4 and mAID-Venus-ANAPC4, respectively, in response to the addition of 0.5 mM auxin for mAID-nanobodies and AID-Venus-ANAPC4 or 1 µg/ml tetracycline to induce deGradFP expression. Cells that efficiently degraded Venus-ANAPC4 during 24 h of deGradFP expression are high-lighted by a dotted line. **b** Degradation kinetics of the time-lapse data shown in **a** represented as the mean ± s.d. for mAID-vhhGFP ($n = 9$, −auxin; $n = 16$, +auxin), mAID-Venus-ANAPC4 ($n = 12$, −auxin; $n = 20$, +auxin), and deGradFP ($n = 8$, −tetracycline; $n = 11$ +tetracycline). **c** Western blot analysis of cells treated as in **a** for the indicated time. Increased levels of CCNB1 are indicative of a mitotic arrest and thus functional inactivation of the APC/C. See also Supplementary Figure 4 for characterization of the mAID-Venus-ANAPC4 knock-in, the effect of prolonged expression of deGradFP on Venus-ANAPC4 and CCNB1 levels, and Venus-ANAPC4 fluorescence in response to expressing deGradFP and deGradFP$^{\Delta Fbox}$

mAID-Venus-ANAPC4 exhibited an almost identical migration behavior in SDS-PAGE as determined by Western blot analysis (Supplementary Figure 4b). Since mAID-Venus-ANAPC4 properly incorporated into the APC/C and neither obvious growth defects were observed during two months of cell culture nor a consistent increase in the duration of mitosis (Supplementary Figure 4e), we conclude that AID-Venus-ANAPC4 cells are suitable to compare the ability of mAID to degrade ANAPC4 and inactivate the APC/C side by side with mAID-nanobody. In addition, we assessed how the mAID-nanobody approach relates to deGradFP as an alternative nanobody degradation technology[8]. To this end, we created isogenic tetracycline-inducible cell lines expressing deGradFP or deGradFP$^{\Delta Fbox}$ as a negative control in the background of the endogenous Venus-ANAPC4 expressing cell line described earlier.

To assess the efficiency of all three systems, we simultaneously monitored Venus-ANAPC4 and mAID-Venus-ANAPC4 fluorescence in single cells by time-lapse microscopy for 24 h, either after the addition of auxin (mAID-nanobody and mAID-Venus-ANAPC4) or after tetracycline-induced deGradFP expression. As previously observed, Venus-ANAPC4 was efficiently targeted for degradation by the mAID-nanobody within 3 h of auxin addition (Fig. 5a). At this time point neither cells with mAID-Venus-ANAPC4 nor cells with Venus-ANAPC4 targeted by deGradFP showed a prominent decrease in the fluorescence signal (Fig. 5a, Supplementary Movie 2).

Even 24 h after induction of degradation, no decrease of mAID-Venus-ANAPC4, both by live cell imaging and by Western blot analysis of ANAPC4 levels in total cell lysates was observed (Fig. 5b, c). To ensure that this result was not caused by

a suppressor mutation within the SCF$^{TIR1}$—auxin system in the two clones we generated (Supplementary Figure 4), we rescued the mAID-Venus-ANAPC4 degradation with the mAID-nanobody. Indeed, expression of mAID-nanobodies resulted in efficient degradation of mAID-Venus-ANAPC4 in response to auxin in transiently transfected cells (Supplementary Figure 5a, Supplementary Movie 3). Of note, degradation occurred in cells containing a wide range of mAID-nanobody transcripts, indicative of a robust degradation system. In agreement, we observed auxin-dependent degradation of mAID-Venus-ANAPC4 in the two clones tested by Western blot analysis of the total cell population (Supplementary Figure 5b). We hypothesize that when directly fused to Venus-ANAPC4 in context of the large APC/C assembly, the mAID degron is not accessible for recognition by the SCF$^{TIR}$ complex. Binding of the mAID-nanobody to Venus might bridge the interaction and recruit SCF$^{TIR}$ to mAID-Venus-ANAPC4.

In addition, the alternative nanobody-based approach deGradFP but not deGradFP$^{\Delta Fbox}$ targeted Venus-ANAPC4 for degradation. However, in contrast to the mAID-nanobody, the degradation kinetics of deGradFP were slower and more heterogeneous (Supplementary Movie 2) and reduced the Venus-ANAPC4 fluorescence in the total cell population by only ~35% (Fig. 5b and Supplementary Figure 6a, b). The lower overall degradation efficacy of deGradFP might be a consequence of the additional time required for expression and folding of the deGradFP protein. Further, the protein levels of deGradFP but not of the mAID-nanobody were strongly stabilized by proteasome inhibition with MG132 (Supplementary Figure 6c, d), suggesting that deGradFP is ubiquitylated and degraded and therefore might become limiting.

We conclude that from the three evaluated approaches only the mAID-nanobody inactivated the APC/C sufficiently to prevent degradation of mitotic substrates such as CCNB1 and CCNA2 (Figs. 4d and 5c). Hence, at least in the context presented here, the mAID-nanobody is the method of choice to functionally inactivate the APC/C.

**mAID-nanobody-dependent degradation in zebrafish.** Finally, we assessed whether auxin-mediated degradation by the mAID-nanobody is also effective beyond human cell lines. We chose zebrafish as a vertebrate organism, since it is an outstanding animal model for time-lapse imaging approaches and thus would especially benefit from a traceable and reversible protein degradation approach. Previously, anti-GFP nanobody[12] and Halo-Tag[24] degradation systems were used in zebrafish. However, inducible and reversible knockdown strategies are still in their infancies. We first established the concentration of IAA zebrafish embryos tolerated during development. Whereas application of IAA up to 2 mM did not result in any visible developmental defect, 5 mM IAA slightly delayed development (Supplementary Figure 7a). Consistent with previous studies in other model organisms[2,5], we therefore used 0.5 mM IAA or 0.5 mM NAA for all subsequent experiments. To assess if auxin-based protein degradation with the mAID-nanobody is possible in zebrafish, we first used a transgenic line expressing proliferative cell nuclear antigen tagged with GFP (GFP-PCNA). We provided the degradation system by injecting a mix of RNAs expressing mAID-nanobody, TIR1 and the injection tracer H2B-RFP into embryos in a mosaic fashion. We added 0.5 mM IAA at 4–5 h post fertilization (hpf) and monitored GFP-PCNA degradation in retinal neuroepithelium (Fig. 6a).

GFP-PCNA was robustly degraded at 22 hpf to less than 20% of its original levels, while only negligible protein degradation occurred in the absence of IAA (Fig. 6b). In case of NAA, we

observed even more robust degradation of GFP-PCNA to just 12% of its original levels (Fig. 6c), suggesting NAA as the preferred molecule for triggering the TIR1–AID interaction in zebrafish. Next, we tested the kinetics of GFP-PCNA degradation by adding NAA to embryos at 22 hpf and followed the GFP fluorescence by time-lapse imaging in S phase cells with and without the degradation system (Fig. 6d). A decrease in GFP signal was apparent from 60 min after NAA addition and after 3 h GFP-PCNA reached low levels that were comparable to long-term incubation of embryos with IAA or NAA for 18 h (Fig. 6b–d). We observed some differences in degradation efficacy among cells (Fig. 6d and Supplementary Figure 7b) most likely reflecting that differing amounts of RNAs are present in individual cells due to differential distribution upon cell division.

To investigate if degradation is also possible in other cellular compartments, we used transgenic lines expressing GFP fused to (i) the Lamina-associated polypeptide 2 (GFP-LAP2b), an inner nuclear membrane protein; (ii) GFP-Utrophin, an F-actin binding probe that is recruited predominantly to the cell cortex and (iii) Myosin, light chain 12, genome duplicate 1-GFP (Myl12.1-GFP), a light chain of non-muscle myosin II that colocalizes with actin filaments in the cytoplasm. In all cases, adding IAA resulted in robust protein degradation (Fig. 6e). Quantification of GFP-LAP2b reiterated that NAA was more potent than IAA for protein depletion in the zebrafish (Fig. 6f). Finally, we wondered whether GFP-tagged proteins could be recognized by the AID-nanobody during or shortly after translation. Thus, we tested whether proteins localized to compartments without SCF$^{TIR1}$ and proteasomal activity such as mitochondria might be subject to mAID-nanobody-driven protein depletion using a zebrafish line expressing GFP fused to a mitochondrial matrix localization sequence (mls-GFP). Indeed, we observed a reduction of GFP fluorescence after IAA addition (Fig. 6g), suggesting that mls-GFP is targeted for degradation immediately after translation and before it is translocated into mitochondria. Since mls does not contain any lysine residues, this result underlines that mAID-nanobodies allow GFP-fusion protein degradation exclusively based on GFP.

Taken together, we demonstrate that the auxin system can be applied to a vertebrate model organism thus is applicable beyond human cell lines. By using zebrafish as a proof of principle, we show that the mAID-nanobody can target GFP-tagged proteins localized to different subcellular structures for degradation.

## Discussion

Instant but reversible inactivation of proteins is key to reveal acute phenotypes that otherwise might be masked by compensation or a dominant terminal phenotype. This holds especially true for proteins or protein complexes such as the APC/C that are essential for proliferation, but fulfill beyond this canonical role additional crucial functions, i.e., during development or in quiescent and post-mitotic cells[25–27]. Therefore, several strategies have been developed that employ the ubiquitin proteasome system present in all eukaryotic cells to target proteins for degradation, either by the addition of cell-permeable molecules[2,28], the expression of genetically encoded nanobodies[8,11,12] or more recently, by the injection of antibodies into human cells or mouse oocytes[23]. All of these approaches have advantages and disadvantages and their successful application depends on the targeted protein(s) and the experimental model of choice.

Here, we present a mAID-nanobody to degrade GFP-tagged proteins that combines advantages of existing auxin and nanobody-based technologies. Utilizing SCF$^{TIR1}$, the mAID-nanobody employs the same intracellular degradation machinery established by the classical auxin system and thus shares its key

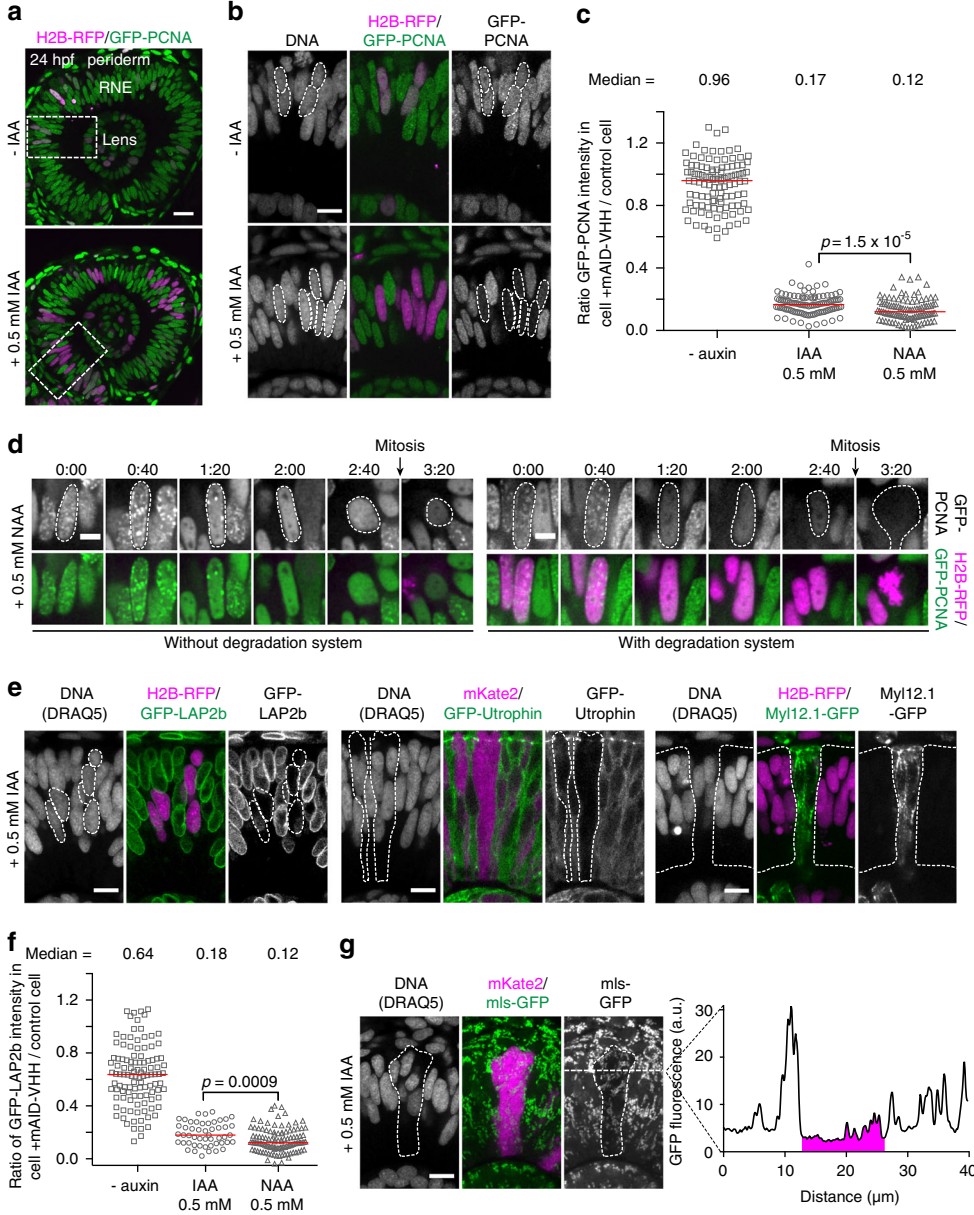

**Fig. 6** Auxin-dependent degradation of GFP-tagged proteins by mAID-nanobodies in zebrafish. **a** Eye of a 24 hpf zebrafish embryo in *Tg(bactin:GFP-PCNA)* mosaically injected with RNAs of TIR1, mAID-nanobody and H2B-RFP RNA to visualize affected cells. The dashed box shows the area displayed in **b**. RNE, retinal neuroepithelium. Scale bar: 20 µm. **b** GFP-PCNA degradation in the presence of IAA. Dashed line: Nuclei of cells expressing the degradation system. Scale bar: 10 µm. **c** Comparison of GFP-PCNA degradation at 22 hpf in response to IAA and NAA treatment. 1 represents no degradation and 0 represents complete degradation. Bars represent median, −auxin control (n = 11, 110 cells), IAA (n = 11, 108 cells), NAA (n = 11, 108 cells). Significance according to a two-tailed Mann–Whitney test. **d** Time-lapse imaging of GFP-PCNA degradation in response to NAA in cells without (H2B-RFP negative, left panel) and with degradation system (H2B-RFP positive, right panel). Scale bar: 5 µm. **e** Degradation of GFP-fusion proteins localized to different cellular structures. Nuclear membrane-localized Lamina-associated polypeptide 2 *Tg(bactin:GFP-LAP2b)*, filamentous actin binding Utrophin fragment *Tg(bactin:GFP-Utrophin)* and actomyosin-associated non-muscle myosin II light chain *Tg(bactin:Myl12.1-GFP)*. Cells containing the degradation system were visualized by cytoplasmic H2B-RFP or cytoplasmic mKate2 RNA co-injection. Scale bar: 10 µm. **f** Comparison of GFP-LAP2b degradation in response to IAA and NAA treatment as in **c**. Bars represent median, −auxin control (n = 11, 110 cells), IAA (n = 5, 50 cells), NAA (n = 11, 111 cells). Significance according to a two-tailed Mann–Whitney test. **g** Depletion of GFP from mitochondria, *Tg(Eef1a1:mls-GFP)*. Cells containing the degradation system show cytoplasmic mKate2 signal. GFP intensity profile along dashed line is plotted, magenta corresponds to the degradation area. Scale bar: 10 µm. All data shown in this figure are representative examples from at least two independent biological experiments

features:[2,3] (i) robust conditional and rapid degradation of proteins localized to distinct compartments: subcellular and cellular structures including chromatin, the nuclear envelope, the cell cortex and cytoplasmic actin filaments (Figs. 2–4 and 6), (ii) depletion of membrane proteins and proteins in context of large macromolecular assemblies such as the nuclear lamina or the APC/C (Figs. 3, 4, and 6), (iii) reversibility after auxin removal

and functional recovery of the targeted protein (Figs. 3 and 4), and (iv) applicability in human tissue or cell culture and in a vertebrate model organism (Figs. 2–6).

By directing SCF$^{TIR}$ activity towards the GFP, the mAID-nanobody can in principle target any protein for degradation regardless of their sequence, size and structure. Even a protein localized to a compartment with no SCF activity such as

mitochondria can be degraded (Fig. 6), presumably by co-translationally binding of the mAID-nanobody to the target once the GFP epitope is folded. Degradation of the model substrate Venus indicated that the classical auxin system enables faster degradation kinetics than the mAID-nanobody (Fig. 2). In the context of fluorescent fusion proteins, mAID-nanobody depletion achieved half-lives ranging from ~16–83 min (Figs. 3 and 4), which we expect to be suitable for most experimental designs.

As the AID degron is provided *in trans* by the nanobody our technology does not require time-consuming fusion of AID degrons to the protein(s) of choice. Thereby, mAID-nanobodies can take advantage of GFP stock collections that exist for several model organisms[13–15] and the increasing number of homozygous fluorescent knock-ins in cell culture[16–19]. Providing the mAID *in trans* by the nanobody might have further benefits over a direct fusion of the degron sequence to the target protein in certain cases. First, titratable expression levels of the mAID-nanobody provides additional means to control the extent of target degradation. Significantly lower levels of mAID-nanobody than the target partially decreased Venus expression to a steady level that was maintained for at least 7 h after auxin addition (Fig. 2). This feature might facilitate establishing thresholds at which the concentration of the target protein becomes limiting for a specific biological function. Second, previous studies raised concerns that AID fusion can destabilize substrate proteins[4], and we observed that fusing mAID to Venus or Venus-ANAPC4 partially altered the localization of the fusion protein. Although adding the mAID degron to Venus is not predicted to hinder free diffusion of the 37 kDa fusion protein through nuclear pore complexes[29], mAID-Venus was localized more pronouncedly to the cytoplasm than Venus (Fig. 2). Similarly, mAID-Venus-ANAPC4 was enriched in the cytoplasm compared to Venus-ANAPC4 and untagged ANAPC4 (Figs. 4 and 5, Supplementary Figures 3 and 4). In contrast, the presence of the mAID-nanobody had neither an effect on the localization of Venus nor of Venus-ANAPC4 (Figs. 2 and 4), nor affected it the viability of Venus-ANAPC4 expressing cells (Supplementary Figure 3). Third, recruiting SCF$^{TIR1}$ by the mAID-nanobody might be advantageous for degradation of subunits in large protein complexes such as the APC/C, where a directly fused AID degron might not be accessible to the SCF$^{TIR1}$ complex (Fig. 5 and Supplementary Figure 5).

Compared to existing anti-GFP nanobody degradation strategies based on deGradFP[8,11,12], the major advantages of the mAID-nanobody is the ability to rapidly induce the degradation of the target protein but also to allow recovery after auxin removal. Further, the stabilization of the mAID-nanobody by lysine mutagenesis ensures that the amount of nanobody does not become limiting as it is the case for antibody-based degradation[23] and likely deGradFP in our experimental system (Supplementary Figure 6). By virtue of endogenously tagged Venus-ANAPC4, we show that mAID-nanobodies rapidly decrease Venus-ANAPC4 levels to an extent that causes a pronounced mitotic arrest. In contrast, even extended expression of deGradFP for 24 h did not reduce Venus-ANAPC4 levels by more than ~35% and failed to cause a mitotic phenotype (Fig. 5). We observed that deGradFP is strongly stabilized by proteasome inhibition (Supplementary Figure 6). Hence, similar to the first version of mAID-vhhGFP4 with lysine residues (Supplementary Figure 1), the vhhGFP4 nanobody in deGradFP might be ubiquitylated and degraded. At least with the tetracyline-inducible system we used here, the achieved degree of deGradFP expression might not be sufficient to completely deplete Venus-ANAPC4.

By establishing mAID-nanobody degradation in zebrafish we demonstrate that the auxin system can be applied to a vertebrate model organism. As observed in human cells efficient degradation in zebrafish might require optimization of mAID-nanobody

expression with respect to the abundance of targeted protein, which can be easily achieved by injecting different amounts of RNA. Because injected RNAs are getting depleted during animal development RNA-based delivery of the mAID-nanobody system is restricted in time. In the retina, we have observed efficient degradation up to 26 h post injection. Creation of a transgenic line expressing TIR1 and an inducible mAID-nanobody would greatly augment applications in zebrafish and other animal models. Since the auxin system has been established in yeast[2,4], flies[1], and nematodes[5], we predict that mAID-nanobody technology can readily be implemented in a wide range of animal models.

In summary, we present an approach in which ectopic expression of TIR1 and a customized mAID-vhhGFP fusion provide a powerful strategy for the conditional and reversible degradation of potentially any GFP/GFP-like fusion protein. mAID-nanobodies enable loss of function studies of endogenous GFP-tagged proteins and thus are especially attractive to animal models where GFP-trap stock collections exist and homozygosity of alleles can be easily achieved by breeding. Facilitated by advances in genome engineering, the number of endogenously GFP-tagged lines in different model organisms and in human cell culture models will only increase in the future. mAID-nanobody degradation may also be applied to overexpressed GFP-fusions, e.g., to inactivate dominant negative mutants. Because knockouts are comparably easier to obtain than knock-in's, the existing GFP BAC libraries in human cells[30] provide an additional resource for loss of function studies by mAID-nanobodies, given that endogenous alleles are ablated by CRISPR/Cas9 or alternative methods. Notably, our approach is not restricted to GFP and GFP-like proteins as nanobodies directly recognizing endogenous proteins[31] or other fluorescent proteins are available[32]. Finally, we present a rationale of how nanobodies in general can mediate ligand-inducible protein interactions without the need of genetically modifying the target of interest, thereby enabling applications beyond auxin-dependent protein degradation.

## Methods

**Cell culture.** Parent and HeLa FRT/TO cells were a kind gift from Jonathon Pines (ICR, London, UK) and HeLa cells expressing INCENP-GFP (INCENP-LAP) from a BAC were a kind gift from Anthony Hyman (MPI-CBG, Dresden, Germany). Cells were cultured according to standard mammalian tissue culture protocols including testing for mycoplasma. HeLa cells were maintained in Advanced DMEM (Gibco) supplemented with 2% FBS, 1% (v/v) penicillin-streptomycin, 1% (v/v) Glutamax and 0.5 μg/ml Amphotericin B. Stable integrants were selected with 400 μg/ml neomycin (AID containing constructs) and 0.5 μg/ml puromycin (TIR1 and free GFP constructs) or 200 μg/ml hygromycin (Venus, Venus-LMNA and CCND1-Venus transgenes).

**DNA constructs.** For mammalian expression we used codon-optimized rice TIR1[6]. For creating TIR1 expressing construct a cassette of N-terminal nuclear localization signal (NLS)-TIR1–C-terminal HA-tag and N-terminal Myc-tag–nuclear export signal (NES HIV rev)–TIR1 separated by a P2A site has been inserted into a Rosa26-CAG-IRES-puro targeting vector upstream of IRES. For generating AID-Venus fusion constructs we used codon optimized *A. thaliana* IAA17 degron (AID) for mammalian expression[6] and N-terminal triple HA-tag–Venus-AID (full length) cassette or N-terminal triple HA-tag–Venus-triple minimized AID (3x mAID) or N-terminal triple HA-tag–Venus-single minimized AID (mAID) has been inserted into the pcDNA5/FRT/TO vector (Thermo Fisher Scientific) for tetracycline inducible overexpression. The vhhGFP4 nanobody-containing tetracycline-inducible overexpression constructs have been generated by insertion of either N-terminal triple HA-tag–mAID–vhhGFP4 or N-terminal triple HA-tag–mAID$^{K→R}$-vhhGFP4 or N-terminal triple HA-tag–mAID$^{K→R}$-vhhGFP4$^{K→R}$ into pcDNA5/FRT/TO vector (Thermo Fisher Scientific).

Sequences: mAID 69–132aa of *A. thaliana* IAA17 (ACPKDPAKPPAKAQVVG WPPVRSYRKNVMVSCQKSSGGPEAAAFVKVSMDGAPYLRKIDLRMYK), mAID$^{K→R}$ (ACPRDPARPPARAQVVGWPPVRSYRRNVMVSCQRSSGGPEAA AFVRVSMDGAPYLRRIDLRMYR), vhhGFP4$^{K→R}$ (MDQVQLVESGGALVQPGG SLRLSCAASGFPVNRYSMRWYRQAPGEREWVAGMSSAGDRSSYEDSVRGR FTISRDDARNTVYLQMNSLRPEDTAVYYCNVNVGFEYWGQGTQVTVSS).

To visualize mAID-vhhGFP expression in transient transfection experiments the mAID-vhhGFP sequence was cloned downstream of an IRES element of CMV-driven mCherry fused at its N terminus to the N-terminal importin β binding domain (IBB) creating IBB-mCherry-IRES-mAID-vhhGFP in a pIRESpuro backbone (Thermo Fisher Scientific). LMNA was amplified from a plasmid encoding GFP-Lamin A[33] and cloned onto the C-terminus of 3xHA-Venus. Flag-murine CCND1 was amplified from a CCND1-CDK2 fusion[34] and cloned onto the N terminus of a 3xHA-Venus-6x histidine tag. Note, residues T286 and T288 were changed to alanine by side-directed mutagenesis. All reporter constructs including Venus (3xHA-Venus-6x histidine tag) and Venus-mAID (3xHA-Venus-mAID) were cloned into a pIREShygro3 backbone (Thermo Fisher Scientific). Sequences encoding deGradFP and deGradFP$^{ΔFbox}$ were a gift from Markus Affolter (Addgene plasmids #35579 and #35580)[8] and subcloned for tetracycline-inducible expression into pCDNA5 FRT/TO (Thermo Fisher Scientific) using EcoRI and XbaI restriction sites.

To target the ANAPC4 gene at the N-terminus with 3xFlag-SBP-Venus ~1,500 base pair long homology arms were amplified from RPE-1 (hTERT) genomic DNA using:

Not I 5′-ATAAGAATGCGGCCGCctttttcatctccccgttgtcgctgaagcc-3′ together with

XhoI 5′-ccgctcgagCAACATGGGGACGGCCctgcaacgacacccagtcagaggccggc-3′ for the left, and

HindIII 5′-cccaagcttCGTTTTCCGACCTGTTTCCCATCCTTCCGGGTG-3′ together with

NotI 5′-ataagaatGCGGCCGCaggtcatctctacaacactcaactcc-3′ oligonucleotides for the right homology arm, respectively. An XhoI-3xFlag-SBP-Venus-15-alanine-glycine-linker-HindIII fragment was amplified from synthetic DNA and cloned together with both homology arms into the NotI sites of pAAV-MCS (Stratagene). For mAID-Venus-ANAPC4 targeting the sequence encoding 3xFlag-SBP was exchanged with mAID using the EcoRI and XhoI sites present in the 3xFlag-SBP-Venus nucleotide sequence.

To generate zebrafish constructs Gateway cloning (Thermo Fisher Scientific) based on the Tol2 kit[35] was used. Tol2-bactin:GFP-PCNA, eGFP-PCNA middle entry clone[36] was combined with the beta-actin promoter 5′ entry clone[35] and pTol2 + pA R4-R2 backbone[37]. Tol2-bactin:GFP-LAP2b, the LAP2b 3′ entry clone was created by amplifying Lap2b from pmRFP_LAP2beta_IRES_puro2b (gift from Daniel Gerlich, Addgene plasmid #21047)[38] using forward 5′-ggggacagctttcttgta caaagtggctCTGTACAAGTACTCAGATCTCGAGC-3′ together with reverse 5′-ggggacaactttgtataataaagttgcTCAGTTGGATATTTTAGTATCTTGAAGAAA ATTAGTG-3′ oligonucleotides) followed by recombination with the beta-actin promoter 5′ entry clone, GFP middle entry clone, and pTol2 + pA R4-R3 backbone[35]. The HA-mAID$^{K→R}$-vhhGFP4$^{K→R}$, Flag-myc-NES-TIR1, NLS-TIR1-HA and mKate2 were subcloned into pCS2+ vector for RNA production. The H2B-RFP in pCS2+ was published previously[39].

### Generation of cell lines.
A Neon Transfection system (Thermo Fisher Scientific) was employed to deliver DNA constructs to generate all cell lines. Stable HeLa FRT/TO and RPE-1 (hTERT) FRT/TO cell lines expressing TIR1, Venus, Venus-LMNA, CCND1-Venus or Venus-AID, mAID-vhhGFP4, deGradFP and deGradFP$^{ΔFbox}$ constructs were cloned into pCDNA5/FRT/TO vectors (see above) and integrated into an FRT site using the Flp-In system (Thermo Fisher Scientific) according to the manufactures instructions, followed by selection with neomycin (400 µg/ml) and isolation of single clones. mAID-vhhGFP4 constructs were inserted into the FRT site of same single parent RPE-1 (hTERT) or HeLa clone expressing already TIR1 to ensure an isogenic background. mAID-nanobodies expression in Venus-LMNA cells was induced for 24 h by 10–20 ng/ml tetracycline and by 1 µg/ml tetracycline for CCND1-Venus or Venus cells unless otherwise indicated. For Venus-ANAPC4 degradation the residual tetracycline in the medium was sufficient for degradation. deGradFP and deGradFP$^{ΔFbox}$ expression was induced by 1 µg/ml tetracycline for the indicated time. Site-specific integration of 3× Flag-SBP-Venus into ANAPC4 loci was facilitated by CRISPR/Cas9-mediated homologous recombination using a gRNA targeting the N-terminus of the ANAPC4 gene (5′-ctgactggggtgtcgttgca-3′). Briefly, NLS-Flag-linker-Cas9[6] and the pAAV-3xFlag-SBP-Venus or pAAV-mAID-Venus targeting constructs (see above) were co-electroplated into HeLa FRT/TO cells followed by one week recovery and expansion of cells. Subsequently, single Venus-positive integrants were identified and sorted by flow cytometry. PCR and sequencing analyses confirmed that in the case of 3xFlag-SBP-Venus-ANAPC4 targeting all ANAPC4 alleles were successfully targeted, whereas in the case of mAID-Venus-ANAPC4 only one out of three ANAPC4 alleles was targeted. A second allele displayed a truncation removing the complete first exon including all potential start codons, while a third allele featured a deletion of 10 base pairs upstream of the ANAPC4 start codon that included the acceptor splice site of the adjacent exon-intron border. Analyses of ANAPC4 levels in total lysates and of ANAPC4 incorporation into the APC/C (Supplementary Figure 4) by Western blot suggested that the tagged allele encodes for the only ANAPC4 protein being expressed.

### Zebrafish husbandry.
TL (RRID:ZIRC_ZL86) of zebrafish was used for outcrossing transgenic lines. Adult fish were maintained and bred at 26 °C. Embryos

were raised at 28.5 °C. Embryos were incubated in E3 medium supplemented with 0.2 mM 1-phenyl-2-thiourea (Sigma-Aldrich P7629) from ~5 hpf to prevent pigmentation. All animal work was performed in accordance with EU directive 2010/63/EU as well as the German Animal Welfare Act, and animal experiments have been approved to comply with all relevant ethical regulations by responsible authorities (Landesdirektion Dresden, Germany).

### Transgenic zebrafish lines.
The GFP-PCNA line Tg(bactin:GFP-PCNA), ubiquitously expressing the GFP-tagged human proliferating cell nuclear antigen (PCNA), and the GFP-LAP2b line Tg(bactin:GFP-LAP2b), ubiquitously expressing the GFP-tagged rat lamina-associated polypeptide 2 were created by injection of 1 nl of the Tol2 plasmid at 30 ng/µl together with the Tol2 RNA at 50 ng/µl into the cytoplasm of one-cell-stage embryos. $F_0$ embryos with fluorescence signal were grown to adulthood, and Tg carriers were identified by outcross with wild-type fish. The GFP-Utrophin Tg(actb2:GFP-Hsa.UTRN)[40], the mls-GFP Tg(XIa.Eef1a1: mlsEGFP)[41] and the myosin II line Tg(actb2:myl12.1-GFP)[42] were published previously.

### Western blotting and APC/C purification.
HeLa cells treated with IAA for the indicated time were washed twice with PBS and directly lysed in LDS sample buffer (Thermo Fisher Scientific). APC/C was purified from asynchronously growing cells using monoclonal ANAPC3 and ANAPC4 antibodies coupled to Dynabeads (Thermo Fisher Scientific) according to the manufactures instructions. Briefly, cells were collected by trypsinization, washed in PBS, resuspended in extraction buffer (30 mM HEPES at pH 7.8, 175 mM NaCl, 2.5 mM MgCl2, 5% glycerol, 1 mM dithiothreitol, 1 mM phenylmethyl sulphonyl fluoride, EDTA-free Protease Inhibitor Cocktail (Sigma-Aldrich, COEDTAF-RO), 0.25% NP-40). After extraction for 20 min on ice, extracts were cleared by centrifugation for 15 min at 16,000×g and the supernatants were used for pulldown and immunoprecipitation experiments. SBP pull-downs were performed with MyOne Streptavidin T1 Dynabeads (Thermo Fisher Scientific) and beads were washed 5 times in extraction buffer before elution with LDS sample buffer (Thermo Fisher Scientific). Proteins were separated by SDS-PAGE using 4–12% Bis-Tris gels in MES or MOPS buffer. Western blotting was performed in 20% methanol/MOPS buffer (all Thermo Fisher Scientific) in a Mini Trans-Blot electrophoretic cell (Bio-Rad), and detected with the following antibodies at the indicated dilutions: ANAPC4 (custom mouse monoclonal antibody, 1:1000, gift from Jonathon Pines, ICR London, UK), ANAPC4 (rabbit polyclonal antibody, 1:2500, Bethyl laboratories, A301–176A), ANAPC3 (custom mouse monoclonal antibody, 1:1000, gift from Jakob Nilsson, The Novo Nordisk Foundation Center for Protein Research, Copenhagen, Denmark), GFP (custom goat polyclonal antibody, 1:5000, protein facility of the Max Planck Institute of Molecular Cell Biology and Genetics, Dresden, Germany), ANAPC10 (custom rabbit polyclonal antibody, 1:2500, gift from Jonathon Pines, ICR London, UK), ANAPC11 (custom mouse monoclonal antibody, 1:1000, gift from Jonathon Pines, ICR London, UK), Cyclin A (rabbit polyclonal antibody, 1:500, Santa Cruz, sc-596, RRID:AB_631330), Cyclin B1 (mouse monoclonal antibody, 1:1000, BD Pharmingen, GNS-1, RRID:AB_395288), GAPDH (rabbit polyclonal antibody, 1:5000, Cell Signaling, 2118), anti-Camelid VHH (rabbit monoclonal antibody, 1:500, GenScript, A01860), ANAPC1 (rabbit polyclonal antibody, 1:2000, Bethyl laboratories, A301–653A, RRID:AB_1210875), ANAPC2 (rabbit monoclonal antibody,1:1000, Cell Signaling, 12301), ANAPC5 (rabbit polyclonal antibody, 1:1000, Bethyl laboratories, A301–026A, RRID:AB_2227119), ANAPC6 (goat polyclonal antibody, 1:500, Santa Cruz, sc-6395, RRID:AB_2074644), ANAPC7 (rabbit polyclonal antibody, 1:1000, Bethyl laboratories, A302–550A, RRID:AB_1998915), ANAPC8 (rabbit polyclonal antibody, 1:1000, Bethyl laboratories, A301–181A, RRID:AB_890562), CDC20 (mouse monoclonal antibody, 1:1000, Santa Cruz, sc-13162, RRID:AB_628089), FZR1 (mouse monoclonal antibody, 1:500, Lab Vision, MS-1116-P, RRID:AB_64192), TUBA1 (mouse monoclonal antibody, 1:5000, Sigma-Aldrich, T5168). mAID-vhhGFP and TIR1 proteins were tagged with the HA epitope (see DNA constructs) and detected with a custom mouse monoclonal anti HA antibody (1:1000). All Western blot experiments were analyzed on a quantitative near-infrared (IR) scanning system (Odyssey, LI-COR) using secondary antibodies coupled to IR-dyes 680RD and 800CW (LI-COR). Uncropped blots of all Western blot data presented in the main figures are shown in Supplementary Figure 8.

### RNA injection into zebrafish.
RNA was synthesized using the SP6 mMessage Machine kit (Thermo Fisher Scientific, AM1340). RNAs were diluted in double-distilled $H_2O$ supplemented with 0.05% phenol red (Sigma-Aldrich, P0290) and injected into one to two cells of 16–64 cell stage embryos to achieve mosaic expression. The injection mixture yielding the most efficient degradation contained 60 ng/µl Flag-myc-NES-TIR1 RNA, 60 ng/µl NLS-TIR1-HA, 4 ng/µl HA-mAID$^{K→R}$-vhhGFP4$^{K→R}$ RNA and 30 ng/µl H2B-RFP or mKate2 RNA to mark the injected cells. The injected volume was 1–2 nl.

### IAA and NAA treatment.
IAA, (indole-3-acetic acid sodium salt, Sigma-Aldrich I5148) was diluted in ddH2O to make 0.5 M stock solution stored at −20 °C and used for 1 month. The NAA (1-naphthaleneacetic acid sodium salt, ChemCruz, sc-296386) was diluted and used identically to IAA and kept at room temperature up

to a month. Both, IAA and NAA were applied in HeLa and RPE-1 (hTERT) cell culture and zebrafish experiments in a final concentration of 0.5 mM.

If not indicated differently, in zebrafish embryos auxin treatment started at 4–5 hpf when embryos were transferred into 90 mm Petri dish containing E3 medium without methylene blue supplemented with 0.5 mM auxin, 5 mM HEPES pH 7.25 to stabilize the pH and 0.2 mM 1-phenyl-2-thiourea. Dechorionation was not necessary, auxin penetrated the chorion efficiently. The embryos were kept in the incubator at 28.5 °C until 22 hpf, when they were dechorionated and fixed. Performing the experiment at 32 °C did not increase the degradation efficiency. The lowest auxin concentration for effective degradation was 100 μM.

**Immunostaining**. To analyze ANAPC4 localization cells grown on coverslips were fixed for 5 min in 4% paraformaldehyde (PFA), permeabilized for 5 min with 1dx buffer (0.1 % Triton X-100, 0,02% Sodium dodecyl sulfate (SDS) in PBS), blocked for 1 h with 2% BSA in PBS and incubated with ANAPC4 antibodies (rabbit polyclonal antibody, 1:1000, Bethyl laboratories, A301–176A) or anti-Flag M2 (mouse monoclonal antibody, 1:10000, Sigma-Aldrich, F3165) for 1 h at room temperature. After extensive washing with PBS-T (0.1% Tween-20 in PBS) secondary antibodies (goat-anti-rabbit Alexa 488, 1:500, Thermo Fisher Scientific, A-11008 or goat-anti-mouse Alexa 594, Thermo Fisher Scientific, A11005) were applied for 1 h in 2% BSA/PBS-T, followed by extensive washing with PBS-T, post-fixation with 4% PFA for 5 min and mounting of cover slides in VECTASHIELD (VECTOR laboratories, H-1000).

Embryos were fixed in 4% PFA in PBS for 4 h at room temperature. Fixed embryos were blocked over night at 4 °C in blocking solution (10% goat serum, Thermo Fisher Scientific, 16210064, 1% BSA, 0.2% Triton X-100 diluted in PBS). Samples were then incubated in blocking solution with 10 μM DRAQ5 (Thermo Fisher Scientific, 62251) for 3 days, shaking at 4 °C. Before imaging, samples were extensively washed in PBS + 0.2% Tween-20.

**Image acquisition**. Living HeLa and RPE-1 (hTERT) cells were imaged using an ImageXpress Micro XLS widefield screening microscope (Molecular Devices) equipped with a 20 × /0.7, air objective (Nikon) and a laser-based autofocus. During the experiment, cells were maintained in a stage incubator at 37 °C in a humidified atmosphere of 5% CO$_2$. All cell lines were grown in 96-well plastic bottom plates (μclear, Greiner Bio-One). Live-cell imaging was performed in a modified DMEM containing 10% (v/v) FBS, 1% (v/v) penicillin-streptomycin, 1% (v/v) Glutamax, and 0.5 μg/ml Amphotericin B without phenol red and riboflavin to reduce autofluorescence[43]. For labeling of nuclei, live cell fluorogenic DNA labeling probe SiR-Hoechst (Spirochrome) was added at a final concentration of 100 nM to the cell culture medium 2 h prior imaging. Images of the cells were acquired during degradation experiments every 5–10 min or for recovery experiments every 20 min using a Spectra X light engine (Lumencor), and an sCMOS (Andor) camera with binning = 2 or 3, and filters for YFP, Cy5 and Texas Red. ANAPC4 immunostainings were acquired using a Deltavision Core Widefield deconvolution fluorescence microscope (Applied Precision Inc.) equipped with Olympus UPlanSApo 100×/1.4 or 60×/1.2 oil immersion lenses, a CoolSNAPHQ2/HQ2- ICX285 camera, and a LED-light engine (Lumencor).

Zebrafish fixed embryos imaging was performed in the LSM 780 (ZEISS) laser scanning confocal microscope using the LD C-Apochromat 40 × /1.1 water immersion objective (ZEISS). The embryos were mounted on the glass-bottom dishes (MatTek Corporation) into 1% agarose diluted in E3 medium. A z stack of 30–40 μm was acquired with 0.56 or 1 μm steps at room temperature. Time-lapse embryo imaging was performed on a spinning disk confocal setup consisting of the IX71 microscope (Olympus), the scan head CSU-X1 (Yokogawa Electric Corporation), UPLSAPO 60×/1.3 silicon oil objective (Olympus) and Neo sCMOS camera (Andor Technology) controlled by iQ 3.0 software (Andor Technology). The samples were mounted into 0.9% agarose in E3 medium containing 0.1 M Hepes, pH 7.25, and 0.01% MS-222 (Sigma-Aldrich). The dish was filled with E3 medium containing 0.01% MS-222 and 0.2 mM N-phenylthiourea and maintained at 28.5 °C. To reduce potential phototoxicity, only a 20 μm stack with 2 μm steps was taken every 20 min for 4 h with 800 ms exposure time, binning = 2 and low laser powers below 5% of 75 mW.

**GFP/Venus fluorescence quantification**. Venus, Venus-ANAPC4 and AID-Venus-ANAPC4 intensities were extracted using a customized image analysis pipeline in MetaXpress (Molecular Devices). Briefly, acquired images were first flat field corrected, subjected to top-hat filtering and segmented based on SiR-Hoechst (Spirochrome) stained nuclei. Intensities from nuclear and cytoplasmic masks were background corrected and normalized to the intensity at the beginning of the experiment. Quantification of Venus-LMNA, CCND1-Venus and INCENP-GFP was performed manually on flat field and background corrected images using Fiji[44]. In case of INCENP-GFP 11 out of 70 mCherry positive cells showed no degradation and were not included for calculation of average degradation curves.

For the presentation, all images were background-corrected with Fiji with rolling ball radius of 75 pixels. In Fig. 4e, images were smoothed with Fiji replacing each pixel with the average of its 3 × 3 neighborhood. Further, the intensity of DNA SiR-Hoechst staining was individually adjusted for different time-points in Figs. 3, 4 and Supplementary Figures 5 and 6 as SiR-Hoechst staining intensity changes

over time. Zebrafish image quantification was performed in Fiji. The stack was subjected to bleach correction (simple ratio) and then the average intensity of GFP-PCNA fluorescence was measured in 3.3 × 3.3 μm ROI using the ROI manager in a z slice containing the central part of the nucleus. For GFP-LAP2b, a 4 μm line was drawn along the high intensity area at the nuclear envelope and average fluorescence value was measured. The average background fluorescence was subtracted and the ratio of fluorescence in a degradation-system-expressing cell to a neighboring control cell was calculated. Value of 1 represents no degradation and value of 0 represents complete degradation.

**Statistical methods**. Prism 6.0 (Graphpad) was used for statistics and to create graphs. Unless otherwise indicated all data are represented as mean ± s.d. from the indicated number of movies. A two-tailed Mann–Whitney U-test was performed to test for significance of distribution differences. Protein half-life and degradation kinetics were determined by a non-linear regression (one-phase decay) with a least squares (ordinary) fit. All experiments are representative of at least three independent repeats if not otherwise stated. No randomization or blinding was used in this study.

**Data availability**. The authors declare that the data supporting the findings of this study are available within the paper and its supplementary information files. Generated plasmids, cell lines and zebrafish are available from the corresponding authors upon request.

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

## Acknowledgements

J.M. is supported by the German Research Foundation (DFG) (Emmy Noether; MA 5831/1-1) and receives funding from the European Research Council (ERC) under the European Union's Horizon 2020 research and innovation program (grant agreement no. 680042). K.D. received funding from the Maria-Reiche-Program of the TU Dresden. C.N. was supported by the German Research Foundation (DFG) [SFB 655, A25] and the by the Human Frontier Science Program (CDA-00007/2011). We would like to thank Magdalena Gonciarz and Frank Buchholz for critically reading the manuscript. Marija Matejcic, Patricia Ramos and Sylvia Schneider are thanked for creating the transgenic zebrafish lines. We acknowledge support by the Fish and Light Microscopy Facilities of the Biotechnology Center of the TU Dresden and the Max Planck Institute of Molecular Cell Biology and Genetics, and support by the Open Access Publication Funds of the SLUB/TU Dresden.

## Author contributions

Conceptualization, J.M and K.D.; Methodology, K.D., J.I., D.M. and J.M; Investigation, K.D., J.I., C.H. and J.M.; Writing—Original draft, K.D., J.I., C.N. and J.M.; Writing—Revised manuscript, K.D. and J.M; Funding acquisition, K.D., C.N. and J.M.; Supervision, C.N. and J.M.

## Additional information

**Competing interests:** The authors declare no competing interests.

