## [Peer Review File · Nature Communications]

Reviewers' comments:

Reviewer #1 (Remarks to the Author):

GFP tagged proteins can be induced for degradation by expressing a GFP-nanobody-F-box fusion (deGradFP). Because it takes a time for expressing a deGradFP protein, GFP-fused proteins cannot be controlled at a precise timing. On the other hand, an auxin-inducible degron (AID) is good at temporal control, because the responsible F-box protein, Tir1, is activated only when its ligand auxin (such as IAA and NAA) is administrated. The authors combined these two technologies for controlling GFP-fusion by auxin. They introduced Tir and an AID-GFP-nanobody without Lys, latter of which recognizes GFP. Tir1 is activated upon auxin addition. Thus, GFP-fusions, but not the AID-GFP-nanobody (without Lys), are polyubiquitylated for degradation. The authors showed that this technology worked in human cultured cells and zebrafish.

This is an interesting new technology working at the protein level, though the system is more complex than the previous systems because three genetic modifications are required (GFP tagging, and AID-GFP-nanobody and Tir1 expressions). An important contribution of this paper is that it showed that the use of the auxin system for the first time in zebrafish. Many GFP-tagged zebrafish lines are currently available, suggesting that the described technology can be used for the functional analysis of these zebrafish lines.

1. "In contrast, endogenous proteins tagged with the green fluorescent protein (GFP) or GFP-like proteins (e.g. YFP, Venus, Citrine) are readily available in human cells⁷⁻⁹ (page 1)". This sentence is misleading. It is true that the referred papers showed CRISPR/AAV-mediated GFP tagging, but the experiments were mostly done with cells in which a single allele was tagged with GFP. To see the phenotype after depletion, the both alleles must be tagged with GFP. As far as I understand, there is no such collection of human cells available. Therefore, one has to make a homozygous GFP tagged cells for applying this technology. If so, using the original AID technology would be simpler. On the other hand, GFP-tagged zebrafish and Drosophila lines are available. Importantly, they can be crossed to make homozygous lines. Therefore, the described technology can be readily used for functional analysis of GFP-tagged lines of these organisms. I suggest that the authors should tone down the use in human cells and emphasize the use in zebrafish and Drosophila.

2. The key point of this technology is that the AID-GFP-nanobody fusion without Lys redirected the target to its binding protein (Figure S1a), although AID-GFP-nanobody fusion was the original and primary target. To show the specificity, the authors showed that GFP-APC4 associated proteins (APC1 and 3) were not affected (Figure 1d). However, I am still wondering if none of the APC/C components were affected. Structural analyses of APC/C showed that APC4 are located proximity of APC5. The authors should look at the expression level of APC5 and other neighboring subunits after GFP-APC4 depletion.

3. Because this paper describes a new technology, experimental conditions should be described in details. I found that some experimental information was not well documented. The authors introduced AID-GFP-nanobody under the control of tetracyclin-inducible promoter at FRT site in HeLa and RPE1 cells (this can be found only in Material and Methods). For example, there is no information how much of Dox was added and how long AID-GFP-nanobody were induced before IAA addition. It might be helpful for readers to show an illustration of the experimental set up.

4. As the authors described in the introduction, the temporal control is a key feature of this technology. When they used zebrafish, it is not clear for me that how fast the degradation can be induced (Figure 3). Do you have any time-lapse data?

5. "IAA, (indole-3-acetic acid sodium salt, Sigma-Aldrich I5148) was diluted in ddH₂O to make 0.5 M stock solution stored at -20C and used for 1 month (page 12)." Thinking of the structure of IAA,

it is likely not so soluble in water. Are you sure that you can make a 0.5 M stock solution in water? The water solubility of IAA is 1.5 mg/mL (= 85.6 mM)(<http://www.hmdb.ca/metabolites/HMDB00197>).

Reviewer #2 (Remarks to the Author):

In this manuscript the authors describe a modification of a degron technology and its application in cells. They have combined two previously described and fairly well established methods: a) the auxin-induced degron (AID) system and b) the use of anti-GFP nanobodies for targeting an E3 ligase that induces ubiquitination and degradation. The advantage of the system described here - as postulated by the authors - is the inducibility of the system: whereas the nanobody-mediated degradation (deGradFP) relies on the induction of expression of the nanobody for induction of degradation, their new system is induced - like the AID degron - simply by addition of auxin, which brings the E3 ligase into contact with the substrate. Thus, the new system should be as fast as the AID system, but it does not require the integration of the AID tag in cases where a GFP-labeled gene is already available. According to the authors, this allows the system to be widely applied to collections of existing GFP-tagged proteins, thus circumventing the need for laborious CRISPR/Cas9-mediated tagging.

Overall, this manuscript represents a technical rather than a conceptual advance. The authors combine two established techniques to generate a supposedly superior degron technology. They show a couple of selected model applications which demonstrate convincingly the feasibility of their approach. Technically, the experiments are clean, and the manuscript is well written. What I am a bit less convinced of is the practical advantage over existing methods that the new method brings. In addition, the authors make a number of claims that are not fully supported by experimental evidence.

Whether this technology-oriented study is within the scope of Nat. Communications is up to the editorial team to decide. For a purely technological contribution, the advance in technology is somewhat modest, but at the same time there is no strong scientific advance resulting from a clever application of the system either.

My concerns in detail are as follows:

1. The prime advantage of the new method over traditional AID-tagging, as advertised by the authors, is the notion that laborious CRISPR/Cas9-mediated introduction of degron tags can be avoided in cases where GFP-tagged alleles of the desired proteins are available. This would make the technique widely applicable without much effort. However, I would argue that this advantage would only apply in cases where both alleles (in diploid cells) are tagged. To my knowledge, this is rarely the case. Since GFP-tagging is usually used to trace the localisation of proteins in vivo, it is generally sufficient to tag one allele. In model organisms, the use of the nanobody-based degron would therefore require breeding in order to generate homozygously tagged substrates. In cell culture this would be impossible. Hence, I do not consider the advantage of pre-existing GFP collections particularly important.

2. As a second advantage (compared to the deGradFP) they list the faster kinetics due to the inducibility of the system (rather than the reliance on induction of the E3 ligase). At the same time, they do not perform any direct comparisons between the three methods (deGradFP, AID versus their new approach). For example, it is in principle possible that their system delays degradation relative to the AID system because of the need for three components to work together. Degradation could become less effective particularly for low-abundance proteins (or towards the "end" of a degradation phase, where the GFP-nanobody fusion is in excess). A careful, well-controlled direct comparison of the three systems with a couple of representative model substrates would therefore be needed in order to convince the reader of the advantages of this

system.

3. It would be necessary to follow the degradation of the model substrates by western blot against the substrate protein (in cases where this was only done by means of tracing the GFP signal). Given the observation that GFP alone can be degraded in the authors' system, it would be necessary to demonstrate that the system is capable of removing the entire fusion protein; i.e. the authors would need to show that a cleavage of the GFP-tag from the substrate or the accumulation of degradation intermediates that have lost the GFP unit is not observed.

4. In the experiment showing the reversibility of the method, the initial treatment with IAA is only 1 h. The authors have shown that in this time degradation is not complete, and a noticeable change in mitotic index is only observable after many hours of IAA treatment. Thus, if IAA is washed out after 1 h already, many of the cells might not even have experienced APC/C depletion to a relevant degree (depending on their cell cycle stage at the time of depletion). The experiment should therefore be carried out under conditions where IAA treatment is prolonged to a stage where depletion is complete.

Additional points:

5. pg. 1: The term "reversible knockout" is inappropriate in the context of the AID system. "Reversible knockdown" would be more appropriate.

6. pg. 3: it is not really surprising to see ubiquitination of the AID-nanobody fusion construct in cis. In fact, this is what one would expect of the system.

7. The authors should discuss the size of the combined tag (HA-GFP plus the AID-nanobody, which will likely associated permanently with the substrate) for the functionality of the substrate. This size could be a disadvantage compared to the relatively small mAID tag.

Reviewer #3 (Remarks to the Author):

Review: „Conditional control of fluorescent fusion protein degradation by an auxin dependent nanobody“ Daniel et al.

The authors describe a strategy for rapid and reversible degradation of fluorescently labeled proteins within living cells. They employ ectopic expression of (i) the plant F-Box protein Tir1 and (ii) a modified GFP-nanobody devoid of lysine residues fused to a minimized auxin-inducible degron motif (mAID). Both components, when genetically introduced in HeLa cells endogenously expressing mVenus-ANAPC4, induce a significant reduction of the FP-labeled protein upon induction with auxin (IAA) by targeting the nanobody/antigen complex to proteosomal degradation. The auxin induced degradation of the fluorescently labeled antigen was visualized and quantified by time-lapse image analysis and western blot analysis. An auxin dependent degradation of FP fusions in the presence of the GFP-nanobody / Tir 1 components was also shown transiently in zebrafish. The data of this descriptive study are well presented and conclusive. However I have doubts that the current manuscript represents a sufficient advance to merit publication in Nature Communications

Major comments

1) Novelty/applicability.

While the work describe some interesting details how to modify the GFP-nanobody for AID (removal of the lysines), the presented results do not contain major novel findings exceeding the deGradFP system previously reported by Caussin et al., 2012. In this context it is questionable whether the proposed heterologous tripartite system based on the modified GFP-nanobody and the

plant Tir1 exceeds the performance of already established systems and thus have the potential to become broadly applicable. Here the authors apply this system to study degradation in an artificial cellular system (HeLa-cells) stably expressing Tir1. A comprehensive comparison with AID-induced degradation of FP-labeled proteins, AID-mediated degradation of endogenous proteins comprising the mAID motif (Holland et al., 2012) and – most importantly - the deGradFP system (Causinus et al., 2012) would be mandatory. In the light of the CRISPR technology I would further assume that directly adding the mAID to endogenous proteins is better suited to study functional effects of reversible protein depletion and to avoid titration effects derived from transient overexpression of the nanobody and/or Tir 1.

2) Data quality

In support of the statement “to combine the best of auxin and nanobody-based technologies in a universal approach to degrade any GFP-tagged protein in a conditional and reversible manner” the authors should definitely add more experimental data underlining this statement and to show the proposed broad applicability. To this end, additional degradation studies on various FP fusions using the described approach will be necessary. Ideally such studies should include different biological systems (yeast, flies etc) and correspond to FPs located in different subcellular compartments. Most importantly the authors should add additional experiments to demonstrate the reversibility of protein degradation upon removal of auxin as the major advantage of their system. Thus this should be demonstrated on at least two additional FPs with different cellular functions/locations by quantitative time lapse imaging and western blotting.

Minor comments:

What is the expression kinetics of GFP-nanobody-mAID compared to GFP-nanobody-F-Box (deGradFP)?

Page 3, I think it is not surprising that the mAID tagged nanobody is preferably degraded in an Auxin dependent manner.

From their descriptive data, the authors assume, that that the mAID-vhhGFP4 but not the construct is mAID-vhhGFP4 K>R construct is preferably subjected to degradation. Since this is the most important finding, the authors should include additional data (e.g. mass spec analysis) to support their assumption and also to define lysine residues which are targeted for ubiquitination. What is the effect of lysine removal on the intracellular binding properties of the GFP-nanobody (e.g. to be determined by FRAP studies)?

The conclusion that this system is suitable to degrade any GFP tagged protein, which based on the observation that the mAID-vhhGFP4 K>R also induce degradation GFP (is not possible with the deGradFP construct but not have been tested by the authors themselves!!) is far overrated. Far more GFP-fusion of different cellular complexes have to be tested.

What is the cellular effect of expression of Tir1? Controls for expression of Tir1 only are missing.

How efficient is degradation mediated by the GFP-nanobody-mAID? In zebrafish the author report a reduction of GFP-PCNA only to 33%. Could this be increased by injection of more RNA?

How fast is the reversion of AID-induced degradation? Figure 2 c shows images only 9 h after removal of IAA. Quantification of FI (shown in suppl. Figure 2 c) started 5 h after removal of IAA.

Why such long gaps?

Does expression of GFP-nanobody-mAID / Tir1 affects cell viability? Long term studies with stable cell models expressing GFP-nanobody-mAID and Tir1 are missing.

Is there a titration effect regarding the cellular amount of GFP-nanobody-mAID and FP fusion?

Suppl. Figure 1 b: Images of HeLa cells expressing unmodified mVenus are missing

Figure 1 d, 2 b, suppl. Fig 1 e, suppl. Fig 2 a: complete WB data should be added to suppl.

Information.

Point-by-point response to the reviewers

We would like to thank the reviewers for their swift reply and for their comments, which we think were very helpful to further improve our manuscript. We hope that you agree that this substantially revised study is now suitable for publication in Nature Communications.

General remarks

Originally structured according as a *Nature Methods Brief Communication* our manuscript has been written in a very succinct way. We realized that many of the reviewer's questions, particularly concerning technical issues and the advantages of our developed approach, might have been raised because of this very concise format. We have now included in the revised version a substantial amount of additional experimental data to reinforce our findings and are confident that we can highlight the "*distinct practical advantage of your method over deGradFP and the AID system.*"

As requested by the reviewers we have performed a careful side-by-side comparison of our approach, the classical AID system, and deGradFP (see below). However, we note that due to systemic differences, especially between both auxin approaches and deGradFP, a fair comparison is problematic: both auxin-mediated degradation approaches allow the conditional depletion within minutes after auxin addition, whereas the deGradFP protein first has to be expressed and correctly folded before it can target proteins for degradation. Both auxin approaches are easily reversible by removal of auxin, whereas reversibility with deGradFP first requires the elimination of the deGradFP protein. On the other hand, only deGradFP and the mAID-nanobody we present here are able to degrade GFP-tagged proteins without further modification of the targeted protein and thus can take advantage of existing GFP-stock libraries.

The mAID-nanobody takes advantage of the same molecular machinery as the classical auxin system. Thus, it is not expected that it can degrade proteins faster *per se*, especially in the light that mAID-nanobody degradation involves a 3-component system compared the 2-component auxin system as correctly pointed out by the reviewers. While we observed a significantly slower degradation of an isolated Venus protein, degradation by the mAID-nanobody in the context of more biological relevant GFP-fusion proteins provided degradation kinetics that are in the range of the classical 2-component auxin system (e.g. $t_{1/2}$ Venus-ANAPC4 = ~16.19 min, $t_{1/2}$ INCENP-GFP = 20.57 min). By virtue of ANAPC4 degradation and APC/C inactivation, we demonstrate that at least in this experimental setting only the mAID-nanobody is able to efficiently deplete ANAPC4 and thus inactivate the APC/C.

One might argue that adding the AID degron directly to a GFP-tagged or untagged protein is always superior to mAID-nanobody-based degradation. However, as argued before this requires an additional genetic modification step. Further, we provide evidence that at least for ANAPC4 and Venus adding a minimized AID degron affects the localization of both proteins, whereas posttranslational binding of the 11 kDa mAID-nanobody has no such effect. That nanobody binding to proteins might even be advantageous compared to a direct fusion with a protein is in agreement with a recent study by the Söllner and Rothbauer labs, who assessed the properties of chromobodies for labeling proteins in living cells in comparison with a direct fusion to the protein of choice (doi:10.1242/dev.118943).

Thus, while each of three methods has its own advantages and disadvantages, we are convinced that by adding the properties of the auxin system to a deGradFP-like approach, we have created a new powerful tool that is complementary to the existing approaches.

As requested by the reviewers, we have performed a comprehensive comparison of mAID-nanobody, AID, and deGradFP technologies. This included the degradation of Venus as a model substrate. In agreement with our notion in the originally submitted manuscript and the discussion in Caussinus et al., 2012, we did not observe degradation of Venus/GFP by deGradGP as presented in the figure to the reviewers below. In contrast, both the AID system as well as the mAID-nanobody were able to target Venus-mAID and Venus for degradation, respectively (see Figure 2 of the revised manuscript).

Figure to the reviewers:

Time-lapse images showing HA-Venus-6-His fluorescence after induction of deGradFP Δ Fbox or deGradFP expression with 1 μ g/ml tetracycline for the indicated time. Data represent mean \pm s.d. (n=30, two independent experiments).

Reviewer #1 (Remarks to the Author):

GFP tagged proteins can be induced for degradation by expressing a GFP-nanobody-F-box fusion (deGradFP). Because it takes a time for expressing a deGradFP protein, GFP-fused proteins cannot be controlled at a precise timing. On the other hand, an auxin-inducible degron (AID) is good at temporal control, because the responsible F-box protein, Tir1, is activated only when its ligand auxin (such as IAA and NAA) is administrated. The authors combined these two technologies for controlling GFP-fusion by auxin. They introduced Tir and an AID-GFP-nanobody without Lys, latter of which recognizes GFP. Tir1 is activated upon auxin addition. Thus, GFP-fusions, but not the AID-GFP-nanobody (without Lys), are polyubiquitylated for degradation. The authors showed that this technology worked in human cultured cells and zebrafish. This is an interesting new technology working at the protein level, though the system is more complex than the previous systems because three genetic modifications are required (GFP tagging, and AID-GFP-nanobody and Tir1 expressions). An important contribution of this paper is that it showed that the use of the auxin system for the first time in zebrafish. Many GFP-tagged zebrafish lines are currently available, suggesting that the described technology can be used for the functional analysis of these zebrafish lines.

We are very pleased that this Referee finds our new technology interesting and especially useful in context of model organisms where GFP-tagged lines are already available. In our introduction and discussion, we have strengthened this notion and emphasized that we provide here first evidence that the auxin system can be applied to a vertebrate model organism.

1. *"In contrast, endogenous proteins tagged with the green fluorescent protein (GFP) or GFP-like proteins (e.g. YFP, Venus, Citrine) are readily available in human cells⁷⁻⁹ (page 1)". This sentence is misleading. It is true that the referred papers showed CRISPR/AAV-mediated GFP tagging, but the experiments were mostly done with cells in which a single allele was tagged with GFP. To see the phenotype after depletion, the both alleles must be tagged with GFP. As far as I understand, there is no such collection of human cells available. Therefore, one has to make a homozygous GFP tagged cells for applying this technology. If so, using the original AID technology would be simpler. On the other hand, GFP-tagged zebrafish and Drosophila lines are available. Importantly, they can be crossed to make homozygous lines. Therefore, the described technology can be readily used for functional analysis of GFP-tagged lines of these organisms. I suggest that the authors should tone down the use in human cells and emphasize the use in zebrafish and Drosophila.*

1) We agree that homozygous GFP-trap alleles are mainly available or easier to generate in animal models and will likely even increase in the future. We particularly emphasize in the revised manuscript that we consider loss-of-function studies in animals as a major application of our method. However, we also identify important applications of our approach in cell lines. Our newly included data on degradation of Venus-ANAPC4 by mAID-nanobody or deGradFP and degradation of a direct AID-Venus-ANAPC4-fusion highlights that in certain cases previously established nanobody- or auxin-mediated protein degradation methods might not provide an alternative for effective target protein depletion (Figure 5 and Supplementary Figure 4). Due to advances in genome engineering and efforts to monitor the behavior and concentrations of molecules in single cells by quantitative microscopy it is very likely that also the number of homozygous endogenous GFP-lines in cell culture models will increase. On the other hand, it is unlikely that there will be a coordinated effort to immediately add the AID degron to such approaches providing an opportunity for the mAID-nanobody technology to render already generated GFP-lines sensitive to auxin-mediated degradation.

2) Further, we note that mAID-nanobody degradation may also be applied to overexpressed GFP-fusions and genome-wide GFP BAC (bacterial artificial chromosome) libraries, e.g. to inactivate

dominant negative mutants or study translation kinetics after reversible depletion of the transgene. Because knockouts are comparably easier to obtain than targeted knock-in's, the existing GFP BAC libraries in human cells provide an additional resource for loss-of-function studies by mAID-nanobodies. In this scenario, the endogenous alleles could be ablated by CRISPR/Cas9 or alternative methods, which in many cases is already done to proof the functionality of the BAC GFP-fusion protein.

2. *The key point of this technology is that the AID-GFP-nanobody fusion without Lys redirected the target to its binding protein (Figure S1a), although AID-GFP-nanobody fusion was the original and primary target. To show the specificity, the authors showed that GFP-APC4 associated proteins (APC1 and 3) were not affected (Figure 1d). However, I am still wondering if none of the APC/C components were affected. Structural analyses of APC/C showed that APC4 are located proximity of APC5. The authors should look at the expression level of APC5 and other neighboring subunits after GFP-APC4 depletion.*

In addition to ANAPC1 and ANAPC3, we present data on the stability of most APC/C subunits including the APC/C co-activators, CDC20 and FZR1, after mAID-nanobody-mediated depletion of Venus-ANAPC4. None of the analyzed subunits showed a degree of degradation comparable to Venus-ANAPC4. However, Venus-ANAPC4 depletion partially affected some subunits, in particular ANAPC2, ANAPC10 and ANAPC11. (Supplementary Figure 3h). We further assessed the stability of the APC/C after Venus-ANAPC4 degradation by immunoprecipitation of the unaffected APC/C subunit ANAPC3. We observed a partial disassembly of the APC/C complex upon Venus-ANAPC4 removal (Supplementary Figure 3i). While it is known that disrupting the integrity of large molecular assemblies can influence the stability of individual subunits of the complex, we cannot exclude that targeting SCF^{Tir1} to Venus-ANAPC4 also resulted in ubiquitylation and degradation of neighboring subunits. This point is now discussed in the manuscript.

3. *Because this paper describes a new technology, experimental conditions should be described in details. I found that some experimental information was not well documented. The authors introduced AID-GFP-nanobody under the control of tetracyclin-inducible promoter at FRT site in HeLa and RPE1 cells (this can be found only in Material and Methods). For example, there is no information how much of Dox was added and how long AID-GFP-nanobody were induced before IAA addition. It might be helpful for readers to show an illustration of the experimental set up.*

We thank the reviewer for pointing out missing experimental information. We revised the Methods sections and the information provided to individual experiments accordingly.

4. *As the authors described in the introduction, the temporal control is a key feature of this technology. When they used zebrafish, it is not clear for me that how fast the degradation can be induced (Figure 3). Do you have any time-lapse data?*

This is indeed an important point. We now provide time-lapse data following the degradation of GFP-PCNA over time (Figure 6d, Supplementary Figure 6b). Our data suggest that significant protein degradation can be reached within 3 hours.

5. *"IAA, (indole-3-acetic acid sodium salt, Sigma-Aldrich I5148) was diluted in ddH2O to make 0.5 M stock solution stored at -20C and used for 1 month (page 12)." Thinking of the structure of IAA, it is likely not so soluble in water. Are you sure that you can make a 0.5 M stock solution in water? The water solubility of IAA is 1.5 mg/mL (= 85.6 mM)(<http://www.hmdb.ca/metabolites/HMDB00197>).*

For our study, we used Indole-3-acetic acid **sodium salt**, Sigma-Aldrich I5148, which is soluble in water and thus not the form of IAA <http://www.hmdb.ca/metabolites/HMDB00197> refers to.

Reviewer #2 (Remarks to the Author):

In this manuscript the authors describe a modification of a degron technology and its application in cells. They have combined two previously described and fairly well established methods: a) the auxin-induced degron (AID) system and b) the use of anti-GFP nanobodies for targeting an E3 ligase that induces ubiquitination and degradation. The advantage of the system described here - as postulated by the authors - is the inducibility of the system: whereas the nanobody-mediated degradation (deGradFP) relies on the induction of expression of the nanobody for induction of degradation, their new system is induced - like the AID degron - simply by addition of auxin, which brings the E3 ligase into contact with the substrate. Thus, the new system should be as fast as the AID system, but it does not require the integration of the AID tag in cases where a GFP-labeled gene is already available. According to the authors, this allows the system to be widely applied to collections of existing GFP-tagged proteins, thus circumventing the need for laborious CRISPR/Cas9-mediated tagging.

Overall, this manuscript represents a technical rather than a conceptual advance. The authors combine two established techniques to generate a supposedly superior degron technology. They show a couple of selected model applications which demonstrate convincingly the feasibility of their approach. Technically, the experiments are clean, and the manuscript is well written. What I am a bit less convinced of is the practical advantage over existing methods that the new method brings. In addition, the authors make a number of claims that are not fully supported by experimental evidence.

Whether this technology-oriented study is within the scope of Nat. Communications is up to the editorial team to decide. For a purely technological contribution, the advance in technology is somewhat modest, but at the same time there is no strong scientific advance resulting from a clever application of the system either.

We are very pleased that the reviewer assesses our presented experiments are well executed, well presented, and convincingly demonstrating the feasibility of our approach. As mentioned in the general remarks in detail (see above), we now provide extended experimental evidence and discuss of how the mAID-nanobody system compares to the existing methods and better line out the advantages of our technology.

My concerns in detail are as follows:

1. The prime advantage of the new method over traditional AID-tagging, as advertised by the authors, is the notion that laborious CRISPR/Cas9-mediated introduction of degron tags can be avoided in cases where GFP-tagged alleles of the desired proteins are available. This would make the technique widely applicable without much effort. However, I would argue that this advantage would only apply in cases where both alleles (in diploid cells) are tagged. To my knowledge, this is rarely the case. Since GFP-tagging is usually used to trace the localisation of proteins in vivo, it is generally sufficient to tag one allele. In model organisms, the use of the nanobody-based degron would therefore require breeding in order to generate homozygously tagged substrates. In cell culture this would be impossible. Hence, I do not consider the advantage of pre-existing GFP collections particularly important.

Please see our answer to Reviewer 1, comment #1

2. As a second advantage (compared to the *deGradFP*) they list the faster kinetics due to the inducibility of the system (rather than the reliance on induction of the E3 ligase). At the same time, they do not perform any direct comparisons between the three methods (*deGradFP*, AID versus their new approach). For example, it is in principle possible that their system delays degradation relative to the AID system because of the need for three components to work together. Degradation could become less effective particularly for low-abundance proteins (or towards the "end" of a degradation phase, where the GFP-nanobody fusion is in excess). A careful, well-controlled direct comparison of the three systems with a couple of representative model substrates would therefore be needed in order to convince the reader of the advantages of this system.

1) We initially did not directly compare degradation kinetics of *deGradFP* to the degradation obtained by mAID-nanobodies after auxin addition, because induction of the degradation process of both system happens at different time scales. While degradation by *deGradFP* can be inducible (we have used in our study a Tet on system), induction always depends on the *de novo* expression of the nanobody construct. Once the F-box-nanobody is generated in a sufficient amount, degradation of the target protein will (gradually) start. In our system, the mAID-nanobody is already available at the time auxin is added. As auxin supply rapidly diffuses into cells and we observe degradation of GFP-fusion proteins to 50% in case of Venus-ANAPC4 already after ~16 minutes. In contrast, it takes *DeGradFP* ~12 hours after Tet induction to reach 25% depletion of Venus-ANAPC4 in the same experimental setting (Figure 5). We also knocked-in an mAID-Venus tag into the *ANAPC4* locus to compare the classical auxin system with our mAID-nanobody on the same endogenous protein. While, all our data suggests that mAID-Venus-ANAPC4 is functional (Supplementary Figure 4), the classical auxin system was not able to target mAID-Venus-ANAPC4 for degradation. While we do not know the underlying molecular reason for this finding, we speculate that the mAID-tag might not be accessible for SCF^{TIR1} in context of the large macromolecular APC/C assembly, whereas the small 11 kDA mAID-nanobody might bridge the interaction to SCF^{TIR1} (Figure 5 and Supplementary Figure 4).

2) We now provide data comparing mAID-nanobody (Venus) and classical auxin system degradation (AID-Venus) side-by-side using Venus as a model substrate (Figure 2 a, b; Supplementary Figure 2a) as highlighted and discussed in the general remarks. We also attempted to include *deGradFP* into this comparison. However, in agreement with Caussinus et al., 2012 we found that a HA-Venus-6his Fusion cannot be degraded (see figure to the reviewers).

3) An important point is indeed that degradation in our 3-component approach could become less effective when the lysine-less nanobody fusion is in excess and competes with the target-bound nanobody for SCF^{TIR1}. To address this, we included now experimental data targeting Venus with different amounts of mAID-nanobody for degradation (Figure 2 c, d and Supplementary Figure 2 b). While we find degradation efficiency to be rather stable in the presence of increasing amounts of mAID-nanobody, we can indeed observe the tendency of a reduced efficiency in the presence of very high mAID-nanobody levels compared to target protein levels. Nevertheless, even in the highest concentrations of mAID-nanobody pronounced degradation was possible but required slightly more time.

4) To provide more evidence that the AID-nanobody can target a multitude of proteins in different compartments and cellular structures we present data on the degradation of a total of 10 proteins in human cell culture and zebrafish (Figure 3, Figure 6 d, Supplementary Figure 6b).

3. It would be necessary to follow the degradation of the model substrates by western blot against the substrate protein (in cases where this was only done by means of tracing the GFP signal). Given the observation that GFP alone can be degraded in the authors' system, it would be necessary to demonstrate that the system is capable of removing the entire fusion protein; i.e. the authors would need to show that a cleavage of the GFP-tag from the substrate or the accumulation of degradation intermediates that have lost the GFP unit is not observed.

In the revised manuscript, we included now Western blot analysis of previously and newly provided degradation experiments performed in human cell lines (unmodified Venus targeted by mAID-nanobody and AID-Venus (Figure 2), Venus-ANAPC4 targeted by mAID-nanobody (Figure 4), and deGradFP and AID-Venus-ANAPC4 (Figure 5 and Supplementary Figure 5). All our data show that the mAID-nanobody removes the total fusion protein and does not only degrade the GFP (which somewhat would be surprising because (i) due to their covalent linkage, fusion proteins are normally degraded *in toto* by the proteasome and (ii) the notion that the AID-nanobody likely allows the ubiquitylation of lysines also present in the fusion protein as indicated by the much faster degradation rates obtained for GFP-fusions as compared to Venus alone (Figures 2-4).

We demonstrate complete removal of Venus-ANAPC4 by Western blot analysis after mAID-nanobody degradation. We could not observe distinct intermediate degradation products by polyclonal anti-ANAPC4 or polyclonal anti-GFP detection and thus conclude that the target protein was completely degraded (Supplementary Figure 3 g). Hence, detection of the Venus/GFP signal provides a valid proxy to estimate the degradation efficiency. Here, microscopy analysis is in fact superior to assess maximal degradation kinetics as it does not suffer from a population effect as Western Blot analysis in case of differences in degradation efficiency between single cells exist.

4. In the experiment showing the reversibility of the method, the initial treatment with IAA is only 1 h. The authors have shown that in this time degradation is not complete, and a noticeable change in mitotic index is only observable after many hours of IAA treatment. Thus, if IAA is washed out after 1 h already, many of the cells might not even have experienced APC/C depletion to a relevant degree (depending on their cell cycle stage at the time of depletion). The experiment should therefore be carried out under conditions where IAA treatment is prolonged to a stage where depletion is complete.

1) We replaced the previously presented data with a panel in which 3 hours of degradation preceded Venus-ANAPC4 recovery (Figure 4 e, f). We additionally included Western blot analysis for protein recovery (Figure 4 g).

2) A successful depletion of ANAPC4 leads to interference with APC/C function, which results in mitotic arrest. In asynchronously cultured cells, the time until the individual cells reach the mitotic stage and thus the phenotype becomes apparent naturally differs. Hence, starting from an asynchrony culture the mitotic index always will be strongly influenced by the cell cycle speed of the experimental model. In fact, our data in Figure 4 shows that cells accumulate in mitosis already early after auxin addition. To support this data, we now present the duration of mitosis of cells directly after auxin addition and observe a dramatic lengthening of mitosis indicative of functional APC/C inactivation (Supplementary Figure 3j).

Additional points:

5. pg. 1: The term "reversible knockout" is inappropriate in the context of the AID system. "Reversible knockdown" would be more appropriate.

We agree and have amended the wording.

6. pg. 3: *it is not really surprising to see ubiquitination of the AID-nanobody fusion construct in cis. In fact, this is what one would expect of the system.*

We agree and rewrote the respective paragraph in the revised manuscript. Of note, the fact that the nanobody present in deGradFP appears to be not targeted for degradation and that Caussin et al., 2012 speculated that there might be a size limit for SCF recognition let us originally hypothesize that also a wildtype mAID-nanobody might be protected from degradation.

7. *The authors should discuss the size of the combined tag (HA-GFP plus the AID-nanobody, which will likely associated permanently with the substrate) for the functionality of the substrate. This size could be a disadvantage compared to the relatively small mAID tag.*

The size of the mAID-nanobody is with just 11kDa much smaller than the classical direct AID tag of 25kDa. It has been previously reported that direct targeting by AID can affect destabilize substrate proteins (Morawska, M. & Ulrich, H. D. An expanded tool kit for the auxin-inducible degron system in budding yeast. *Yeast* **30**, 341–351 (2013)) and we observed that fusion of mAID to Venus or Venus-ANAPC4 altered the localization of both proteins. We did not notice such an effect on Venus-ANAPC4 or Venus by the continuous presence of mAID-nanobody (Figure 2a, Supplementary Figure 2a, Figure 4a, Supplementary Figure 4d, see also general remarks above). Thus, *in trans* targeting of substrate proteins could in certain cases even be advantageous. For Venus-ANAPC4 we also tested if the continuous presence of mAID-nanobody affected cell proliferation and did not see any indication of a negative effect (Supplementary Figure 3 b). This suggests that mAID-nanobody binding does not interfere with ANAPC4 function. In agreement, it has also no effect on integration of Venus-ANAPC4 into APC/C and the mitotic functions of APC/C (Supplementary Figure 3i, see t=0; Figure 4e). On the other hand, the mAID-nanobody enabled substantial degradation of Venus-ANAPC4 out of the APC/C complex, while neither the direct mAID-Venus-ANAPC4 fusion nor deGradFP achieved comparable degree of depletion (Figure 5, Supplementary Figure 5).

Reviewer #3 (Remarks to the Author):

Review: „Conditional control of fluorescent fusion protein degradation by an auxin dependent nanobody” Daniel et al.

The authors describe a strategy for rapid and reversible degradation of fluorescently labeled proteins within living cells. They employ ectopic expression of (i) the plant F-Box protein Tir1 and (ii) a modified GFP-nanobody devoid of lysine residues fused to a minimized auxin-inducible degron motif (mAID). Both components, when genetically introduced in HeLa cells endogenously expressing mVenus-ANAPC4, induce a significant reduction of the FP-labeled protein upon induction with auxin (IAA) by targeting the nanobody/antigen complex to proteosomal degradation. The auxin induced degradation of the fluorescently labeled antigen was visualized and quantified by time-lapse image analysis and western blot analysis. An auxin dependent degradation of FP fusions in the presence of the GFP-nanobody / Tir 1 components was also shown transiently in zebrafish. The data of this descriptive study are well presented and conclusive. However I have doubts that the current manuscript represents a sufficient advance to merit publication in Nature Communications

We are very pleased that the reviewer evaluates the data in our manuscript as well presented and conclusive. We appreciate the constructive advices for revision and have addressed the Referee's specific major and minor comments below.

Major comments

1) Novelty/applicability.

While the work describe some interesting details how to modify the GFP-nanobody for AID (removal of the lysines), the presented results do not contain major novel findings exceeding the deGradFP system previously reported by Caussin et al., 2012. In this context it is questionable whether the proposed heterologous tripartite system based on the modified GFP-nanobody and the plant Tir1 exceeds the performance of already established systems and thus have the potential to become broadly applicable. Here the authors apply this system to study degradation in an artificial cellular system (HeLa-cells) stably expressing Tir1. A comprehensive comparison with AID-induced degradation of FP-labeled proteins, AID-mediated degradation of endogenous proteins comprising the mAID motif (Holland et al., 2012) and – most importantly - the deGradFP system (Caussin et al., 2012) would be mandatory. In the light of the CRISPR technology I would further assume that directly adding the mAID to endogenous proteins is better suited to study functional effects of reversible protein depletion and to avoid titration effects derived from transient overexpression of the nanobody and/or Tir 1.

1) Since the classical AID system and our newly developed degradation approach rely on the same auxin and SCF^{TIR1}-mediated protein degradation pathway, we expect that our approach could provide at best the same degradation rates as a direct fusion of AID to the substrate protein (see also general remarks above). We provide in the revised manuscript now degradation kinetics/time-lapse imaging data for mAID-nanobody targeting a multitude of substrates in different cellular localisations in cultured cells as well as in zebrafish. These data provide now broader information about the degradation kinetics and the robustness of a tripartite system mAID-nanobody system (Figure 2, Supplementary Figure 2 ,Figure 3, Figure 6d, Supplementary Figure 6b). Depending on targeted GFP/GFP-like fusion protein, protein degradation rates can be within in the range of what has been described previously for the AID system in mammalian cells (Holland, A. J., Fachinetti, D., Han, J. S. & Cleveland, D. W. Inducible, reversible system for the rapid and complete degradation of proteins in

mammalian cells. *Proceedings of the National Academy of Sciences* **109**, E3350–7 (2012)). For instance, the half-life of Venus-ANAPC4 and INCENP-GFP degradation are in the range of 16-20 minutes and thus as fast as the classical auxin system can degrade Venus-mAID in the same experimental setting (see Figure 2).

2) The direct comparison of degrading Venus by a AID-fusion or the mAID-nanobody targeting shows that for Venus alone the classical AID system (Figure 2 a, b, Supplementary Figure 2a) is faster than the mAID-nanobody. While this might reflect the differences between a 2 versus 3-component system, also the additional lysine present in the mAID degron on Venus (Venus-mAID (wt)) but not in the lysine-less mAID-nanobody might contribute to degradation efficiency. Nevertheless, degradation half-lives of GFP-fusion proteins by mAID-nanobodies in a range from 16-83 minutes should be suitable to address most biological questions.

3) To also experimentally highlight possible benefits of our newly developed degradation approach, we provide in the revised version of our manuscript a direct comparison between the three methods deGradFP, AID, and the mAID-nanobody using endogenous Venus-ANAPC4 as a biological relevant model substrate (Figure 5 and Supplementary Figure 4). We demonstrate that out of the three tested approaches only the mAID-nanobody technology realized a substantial Venus-ANAPC4 depletion leading to APC/C inactivation. This case highlights that ultimately the nature of protein to be targeted and the experimental system employed will strongly influence the choice of the best degradation technology. Hence, alternatives to the established protein depletion approaches are still required.

4) By the example of Venus-ANAPC4 depletion, we show that the *in trans* working mAID-nanobody system might be indeed advantageous in case of a complex macromolecular assembly than a directly fused AID degron (see also our answer to Reviewer 2, comment #2 and the general remarks).

5) We now provide results for stable as well as transient applications of our system in cultured human cells as well as in zebrafish (Figure 2-4, Figure 6). We see effective depletion of target proteins on both, endogenous and overexpressed proteins. As expected, transient application leads to a higher variation in degradation rates, likely affecting the different amounts on mAID-nanobody delivered (Figure 3d). Based on the presence of the fluorescent protein, our approach allows quantifying the degree of depletion on the single cell level and thus a reliable phenotype analysis. Further, different degrees of depletion might be useful to relate the degree of depletion to the emergence of a phenotype. We discuss these results and present more data regarding the titration effects of mAID-nanobody levels in Figure 2d and Supplementary Figure 2b (see also our answer to Reviewer 2, comment #2)

2) Data quality

In support of the statement “to combine the best of auxin and nanobody-based technologies in a universal approach to degrade any GFP-tagged protein in a conditional and reversible manner” the authors should definitely add more experimental data underlining this statement and to show the proposed broad applicability. To this end, additional degradation studies on various FP fusions using the described approach will be necessary. Ideally such studies should include different biological systems (yeast, flies etc) and correspond to FPs located in different subcellular compartments. Most importantly the authors should add additional experiments to demonstrate the reversibility of protein degradation upon removal of auxin as the major advantage of their system. Thus this should be demonstrated on at least two additional FPs with different cellular functions/locations by quantitative time lapse imaging and western blotting.

1) We agree with the reviewer and we provide now additional experimental results to demonstrate that mAID-nanobody enables effective substrate degradation from various cellular structures, macromolecular assemblies and compartments - in a total of 10 proteins in human cells and zebrafish. (Figure 2a, b, Figure 3, Figure 4a, Figure 6). In addition, we show the degradation of Venus and Venus-ANAPC4 by Western blot and also recovery in case of Venus-ANAPC4 (Figures 2 and 4). By the example of Venus-ANAPC4 we show that after mAID-nanobody degradation the complete target protein is removed by mAID-nanobodies (Supplementary Figure 3g). Hence, detection of the Venus/GFP signal in general provides a valid and even more sensitive proxy to estimate the degradation efficiency and the recovery. Therefore, we included time-lapse imaging results on protein degradation and recovery of three different GFP/GFP-like fusion proteins localizing to distinct cellular structures (Figure 3d).

2) We do not think that validating our approach in yet another model organism such as yeast or flies would add a significant scientific advance in the context of our study for the following reasons: (i) Our approach takes advantage of exactly the same molecular machinery as the auxin system and all our experiments thus far suggest that degradation by the AID-nanobody shares the key features of the auxin system that have been described in yeast, flies, nematodes and human cells. (ii) The auxin system has already been established in the models the reviewer suggested (yeast and flies) and due to (i), we do not see any indication that the AID-nanobody should not work in these models. (iii) Instead, our study presents first evidence that the auxin system can be applied to a more complex vertebrate model organism (zebrafish) and thus raises possibilities for the auxin system beyond the already established organism.

Minor comments:

What is the expression kinetics of GFP-nanobody-mAID compared to GFP-nanobody-F-Box (deGradFP)?

The mAID-nanobody can be constitutively expressed in our degradation system or induced for a long time before the addition of auxin (in our experiments 24 hours). Thus, the time to express the mAID-nanobody is not a limiting factor as it is for the deGradFP technology. This represents a major advantage in temporal control of our mAID-nanobody degradation approach.

Page 3, I think it is not surprising that the mAID tagged nanobody is preferable degraded in an Auxin dependent manner.

We agree with the reviewer and we revised our writing accordingly.

Form their descriptive data, the authors assume, that that the mAID-vhhGFP4 but not the construct is mAID-vhhGFP4 K>R construct is preferable subjected to degradation. Since this is the most important finding, the authors should include additional data (e.g. mass spec analysis) to support their assumption and also to define lysine residues which are targeted for ubiquitination.

While we agree that the lysine mutagenesis was central for our approach and from an academic point of view the knowledge of which lysine residues in mAID-vhhGFP4 are ubiquitylated by SCF^{TIR1} is interesting. However, we do not believe that this information advances the manuscript or gives important information to a reader, who that might want to apply our method. Precisely, due to the fact that wildtype mAID-vhhGFP4 is ubiquitylated and degraded it is not useful for our approach. Hence, we employed the lysine-less mAID-nanobody that cannot be ubiquitylated .

What is the effect of lysine removal on the intracellular binding properties of the GFP-nanobody (e.g. to be determined by FRAP studies)?

We agree that this is an interesting question, however our results clearly demonstrate that recognition of GFP/GFP-like proteins by the lysine-less GFP-nanobody is still possible and can be very efficient as judged by the fast degradation of Venus-ANAPC4 or INCENP-GFP. In fact, we speculate that the interaction between the nanobody and GFP/Venus is very strong since the lysine-less nanobody to a minor extent is still sensitive to auxin addition. While we cannot exclude that ubiquitylation at the N-terminal methionine is cause for degradation, we favour the idea that some of the nanobody is co-degraded by proteasome with the ubiquitylated substrate. As we show in Supplementary Figure 1, for the functionality of our system there is no alternative to the complete lysine mutagenesis of the nanobody. Hence, while we believe that the intracellular binding properties of the GFP-nanobody are interesting, they do not advance mAID-nanobody-mediated degradation further.

The conclusion that this system is suitable to degrade any GFP tagged protein, which based on the observation that the mAID-vhhGFP4 K>R also induce degradation GFP (is not possible with the deGradFP construct but not have been tested by the authors themselves!!) is far overrated. Far more GFP-fusion of different cellular complexes have to be tested.

1) We agree with the reviewer and we toned down the statement to “mAID-nanobody should **in principle** enable the degradation of any GFP-tagged protein, independently of the availability of additional lysine residues present in the target protein. Notably, this a difference to deGradFP, which cannot degrade GFP as suggested by Caussin et al., 2012 and now confirmed by us (see accompanying Figure to the reviewers). As the data demonstrating that deGradFP cannot degrade an HA-Venus-6xHis protein (the mAID-nanobody can: Figure 2) does not significantly advance the manuscript further, we present this data only here for your consideration.

2) In total, we demonstrate in the revised manuscript the mAID-nanobody-dependent degradation of 10 proteins localized to different compartments, subcellular structures and macromolecular assemblies indicating that our technology can robustly be applied to a multitude of proteins in different experimental models (see also above, answer to major comment #2).

What is the cellular effect of expression of Tir1? Controls for expression of Tir1 only are missing.

The use of TIR1 in animal cells has been established in several previous studies and no pathologic effects have been reported. Overexpression of TIR1 and incorporation could possibly interfere with SCF function. SCF ubiquitin ligases function in cell cycle regulation and proliferation control. Thus, we analysed effect of TIR1 in our experimental system on proliferation rate of cells (overexpression of TIR1 in form of the NLS/NES construct we used in our experimental system). We did not observe any proliferation differences of the TIR1 expressing cell line as compared to the parental Hela cell line. We included this data in the revised manuscript as Supplementary Figure 3b.

Also in the zebrafish, we did not observe noticeably increased cell death or cell cycle abnormalities in cells containing the degradation system compared to surrounding control cells. The cell lines stably expressing TIR1 did not display any growth defect or other phenotype of a prolonged time of cell culture.

How efficient is degradation mediated by the GFP-nanobody-mAID? In zebrafish the author report a reduction of GFP-PCNA only to 33%. Could this be increased by injection of more RNA?

In our experiments, we were already injecting a substantial amount of RNA and increasing the injected RNA further could trigger negative side effects. However, we found other strategies how

to further optimize the experimental protocol and achieved on average 88% degradation of GFP-PCNA and GFP-LAP2b (Figure 6c and f). This was achieved by (i) slight changes in ratios of the injected RNAs, (ii) earlier incubation with auxin (from right after injection ~4 hpf, instead of only after gastrulation ~10 hpf, and (iii) by examining the degradation in embryos 2 hours earlier, at 22 hpf instead of 24 hpf. By this time shift we made sure that the injected RNAs are still abundant enough to facilitate effective target depletion.

How fast is the reversion of AID-induced degradation? Figure 2 c shows images only 9 h after removal of IAA. Quantification of FI (shown in suppl. Figure 2 c) started 5 h after removal of IAA. Why such long gaps?

The reversion of mAID-nanobody degradation depends on the endogenous expression regulation of the target protein (i.e. the speed of translation and protein folding), and thus will differ for every substrate. We analysed now reversibility in time-lapse imaging for three additional GFP-fusion proteins to support the conclusion that our degradation is reversible for target proteins localized to different cellular compartments and subcellular structures (Figure 3). The example of Venus-ANAPC4 recovery allowed us to address also functional recovery of a depleted endogenous protein, which we observe already 7-8hours after auxin removal and thus long before the ANAPC4 levels pre-treatment were reached (Figure 4e). We replaced the previous quantification of recovery with data, which documents recovery following 3 hours of depletion without any time gaps (Figure 4f).

Does expression of GFP-nanobody-mAID / Tir1 affects cell viability? Long term studies with stable cell models expressing GFP-nanobody-mAID and Tir1 are missing.

As mentioned above the use of TIR1 in animal cells has been established in several previous studies and no pathologic effects have been reported. In our proliferation analysis of parent, Venus-ANAPC4, TIR1, and Venus-ANAPC4 + TIR1 + persistent mAID-nanobody expression, we found no evidence that the expression of GFP-nanobody-mAID or Tir1 have adverse effects (Supplementary Figure 3b). Further, we also did not notice any significant proliferation defects during month of cell culture (data not shown).

Is there a titration effect regarding the cellular amount of GFP-nanobody-mAID and FP fusion?

We observed effective degradation once the mAID-nanobody level exceeded or came close to the level of the target protein. However, high amounts of free mAID-nanobody might compete with target-bound mAID-nanobody for recognition by SCF^{TIR1}. Thus, excessive levels of mAID-nanobody might interfere with degradation efficiency. We addressed the titration effects of the system in the newly incorporated Figure 2d and Supplementary Figure 2b and discuss this feature in the main text.

Suppl. Figure 1 b: Images of Hela cells expressing unmodified mVenus are missing

We thank the reviewer for this hint and have replaced the previous figures with a panel containing the missing control.

Figure 1 d, 2 b, suppl. Fig 1 e, suppl. Fig 2 a: complete WB data should be added to suppl. Information.

We have provided complete WB data for Venus-ANAPC4 degradation in Supplementary Figure 3 and are happy to include all scans of all Western blot data as an additional Supplementary Figure when requested by the editor.

Reviewers' comments:

Reviewer #1 (Remarks to the Author):

The authors cleared most of my concerns by showing new data. However, I would like to raise one point that I still do not understand.

The authors compared three technologies (mAID-nanobody, classical AID, and deGradFP systems) by targeting ANAPC4 (Figure 5). They showed that the classic AID did not work at all with mAID-Venus-ANAPC4 (Figure 5a-c), even though it worked well with Venus-mAID (Figure 1). I am wondering what was the reason, because there is no explanation about this observation in the text.

It can be possible that this particular cell line might have a suppressor mutation that blocks auxin-induced degradation. Have you checked multiple clones to confirm the phenotype? If the authors did not see any degradation in the mAID-Venus-ANAPC4 cells, it does not make sense to compare the classical AID with the others in Figure 5. Either showing degradation of mAID-Venus-ANAPC4 or taking out this data is needed.

Apart from the above issue, I feel that the authors did a great work, worth considering for publication in Nature Communications.

Minor points

Page 4, line 10

Supplementary Fig. 1g -> Supplementary Fig. 1e

Page 4, line 13

Fig. 2a, Supplementary Fig. 1a -> Fig. 2a, Supplementary Fig. 2a

Reviewer #2 (Remarks to the Author):

The authors have given the manuscript an extensive overhaul in answering the reviewers' issues, and they have done so in a careful and comprehensive manner. I am now happy with the state of the manuscript and support publication.

Reviewer #3 (Remarks to the Author):

Review: „Conditional control of fluorescent fusion protein degradation by an auxin dependent nanobody“ Daniel et al.

In the revised version the authors have responded comprehensively to some but not to all of my concerns.

The authors now show for more (in summary 10) tested proteins, that auxin-induced mAID-nanobody-mediated degradation of fluorescently-labeled proteins is functional in mammalian cells and zebrafish. However, the requested detailed comparison of the mAID-nanobody approach to direct AID tagging or the deGradFP system lack some important features and control experiments. The authors compared auxin induced degradation of Venus-tagged ANAPC4 mediated by the mAID-nanobody with degradation of mAID-tagged Venus-ANAPC4. In the light that no degradation of mAID-Venus-ANAPC4 construct was observed, the authors concluded that this construct cannot be addressed via the auxin-induced degradation system. As no further explanation is given, this answer is very unsatisfying and put the whole system in question. It would have been far more appropriate to compare the degradation of Venus-tagged ANAPC4 via the mAID-nanobody to degradation of directly mAID-tagged ANAPC4.

For the deGradFP system the authors observed only a very low degradation of Venus-ANAPC4. Although in the text it is stated that an endogenous Venus-ANAPC4 was used, I suppose the authors used their 3xFlag-SBP-Venus tagged version of ANAPC4. Thus, it is conceivable that this construct is not very well recognized by the deGradFP nanobody construct. Further experiments to analyze this possibility (e.g. Co-IPs) would be needed. Furthermore experiments (e.g. WB) which show the expression of the deGradFP are missing. Based on the data as presented I think the comparison between the three systems is not conclusive.

To me a more thorough comparison of the mAID-nanobody approach with the two established techniques (AID, deGradFP) is mandatory to justify the claimed broad applicability and novelty - and consequently - publication of this additional application of the GFP-nanobody in Nature Communications.

Minor comments (but important)

What is the effect of lysine removal on the intracellular binding properties of the GFP-nanobody? The exchange of three lysine residues within a nanobody (115 aa) is quite a substantial modification. Thus a potential effect on binding should be carefully evaluated (comparing to wt nanobody), which become even more important considering the low effect of the deGradFP on Venus-ANAPC4, which might be due to a lower accessibility of the epitope. Such experiments are also important for discussing the advantages of a trans-binding compared to cis-modified constructs, since a high affinity trans-binding nanobody might have dramatic effects on FP-labeled proteins regarding subcellular localization or dynamics (partially demonstrated by the authors themselves showing a cytoplasmic degradation of a GFP protein comprising a mitochondrial targeting sequence). A quite simple option to test intracellular binding would be intracellular IPs of cells co-expressing HA-tagged Venus and mAID-nanobody or mAID(K>R)-nanobody(K>R) using Protein A sepharose (the GFP-nanobody binds to protein A/G sepharose (Rothbauer et al., Mol. Cell. Proteomics 2008)).

Figure 2 c and Figure 2 g)

WB data for mAID-nanobody should be included to visualize a potential effect of auxin treatment / removal on the nanobody.

Figure 3 a /b)

The recovery of Venus-LMNA and CCND1-Venus are very low. After removal of auxin, expression/fluorescence of both proteins only reaches 40 - 50% of their original level. The authors should comment on this.

Supplementary Figure 2 b)

How quantification of mAID-vhhGFP levels was performed? There is no explanation in the figure legends or in the Mat&Meths section.

Supplementary Figure 3 e/f

What was the basis for WB quantification? How many biological/technical replicates? There is no information in the figure legends / Mat&Meths section.

Discussion (page 12)

What do the authors mean by "the small 11 kDa mAID-nanobody"? The fusion of the GFP-nanobody (13 kDa) to a HA-tag (1 kDa) and the mAID (7.5 kDa) results in a protein with a molecular weight of ~ 21.5 kDa (as shown by the authors in WB analysis (Suppl. Figure 1 e)). Since this statement is not correct, the advantages compared to the larger deGradFP (34 kDa) are somehow limited and have to be discussed more properly.

Mat&Meth

Do HeLa cells comprising 3 ANAPC4 alleles?

Reference #9

This ref. does not refer to the expression of nanobodies in living cells.

References #16 - #18

None of those publications refer to GFP-stock collections for cell culture models. E.g. Ratz et al, show mostly CRISPR-mediated heterozygous tagging of various proteins with GFP.

Point-by-point response to the reviewers

We would like to thank the reviewers for their comments. We followed their suggestion of adding additional controls to further strengthen the manuscript. These further changes we believe should now make the manuscript suitable for publication in Nature Communications.

Reviewer #1 (Remarks to the Author):

The authors cleared most of my concerns by showing new data. However, I would like to raise one point that I still do not understand.

The authors compared three technologies (mAID-nanobody, classical AID, and deGradFP systems) by targeting ANAPC4 (Figure 5). They showed that the classic AID did not work at all with mAID-Venus-ANAPC4 (Figure 5a-c), even though it worked well with Venus-mAID (Figure 1). I am wondering what was the reason, because there is no explanation about this observation in the text.

It can be possible that this particular cell line might have a suppressor mutation that blocks auxin-induced degradation. Have you checked multiple clones to confirm the phenotype? If the authors did not see any degradation in the mAID-Venus-ANAPC4 cells, it does not make sense to compare the classical AID with the others in Figure 5. Either showing degradation of mAID-Venus-ANAPC4 or taking out this data is needed.

The reviewer made a very valid point here that has now been addressed. We used the two previously generated mAID-Venus-ANAPC4 clones (described in Supplementary Fig. 4) that both showed no degradation as highlighted by clone #1 shown in Fig. 5a, b. To exclude that a suppressor mutation in the SCF^{TIR1} - auxin system prevents the mAID-Venus-ANAPC4 degradation as suggested by the reviewer, we have transiently expressed the mAID-nanobody in both mAID-Venus-ANAPC4 clones.

Importantly, cells that are positive (mCherry +) for the mAID-nanobody now efficiently degrade mAID-Venus-ANAPC4 upon auxin supply (new Supplementary Fig. 5 and Supplementary Movie 3), demonstrating that the SCF^{TIR1} - auxin system is unperturbed in both clones. We speculate that in the context of the large APC/C assembly, the mAID degron cannot be recognized by SCF^{TIR1}. By binding to Venus the mAID-nanobody might bridge the interaction between SCF^{TIR1} and mAID-Venus-ANAPC4. We mention this hypothesis now in the results and discussion part.

Due to this result, we think the mAID-Venus-ANAPC4 data in Figure 5 is valuable to the reader as it provides a case example, where degradation by an AID degron provided in trans via nanobody binding to the targeted protein is advantageous to a direct AID fusion.

Apart from the above issue, I feel that the authors did a great work, worth considering for publication in Nature Communications.

We thank the reviewer for this positive recommendation.

Minor points

Page 4, line 10

Supplementary Fig. 1g -> Supplementary Fig. 1e

This has been changes accordingly

Page 4, line 13

Fig. 2a, Supplementary Fig. 1a -> Fig. 2a, Supplementary Fig. 2a

This has been changes accordingly

Reviewer #2 (Remarks to the Author):

The authors have given the manuscript an extensive overhaul in answering the reviewers' issues, and they have done so in a careful and comprehensive manner. I am now happy with the state of the manuscript and support publication.

We thank the reviewer for this positive recommendation.

Reviewer #3 (Remarks to the Author):

Review: „Conditional control of fluorescent fusion protein degradation by an auxin dependent nanobody” Daniel et al.

In the revised version the authors have responded comprehensively to some but not to all of my concerns.

The authors now show for more (in summary 10) tested proteins, that auxin-induced mAID-nanobody-mediated degradation of fluorescently-labeled proteins is functional in mammalian cells and zebrafish. However, the requested detailed comparison of the mAID-nanobody approach to direct AID tagging or the deGradFP system lack some important features and control experiments. The authors compared auxin induced degradation of Venus-tagged ANAPC4 mediated by the mAID-nanobody with degradation of mAID-tagged Venus-ANAPC4. In the light that no degradation of mAID-Venus-ANAPC4 construct was observed, the authors concluded that this construct cannot be addressed via the auxin-induced degradation system. As no further explanation is given, this answer is very unsatisfying and put the whole system in question. It would have been far more appropriate to compare the degradation of Venus-tagged ANAPC4 via the mAID-nanobody to degradation of directly mAID-tagged ANAPC4.

- 1) Please see also our detailed response to reviewer 1. In the revised manuscript, we directly compared the efficiency and degradation kinetics of both auxin systems using the simplest model protein for fluorescent and Western analyses - GFP/Venus- in Figure 2. We did not include deGradFP here, as in agreement with Caussin et al., we found that deGradFP cannot target free GFP/Venus for degradation (see Figure to the reviewers, in our previous response to reviewers). Therefore, we have performed the comparison with deGradFP in context of the Venus-ANAPC4 now shown in Figure 5.
- 2) We now provide compelling evidence that mAID-Venus-ANAPC4 can be targeted by the auxin system in the two different knock-in clones (new Supplementary Fig. 5), namely by transfection of the mAID-nanobody. Thus, while in the case of ANAPC4 a direct fusion with AID is not the method of choice for auxin-dependent degradation it provides an excellent example where application of the mAID-nanobody is advantageous.

We hypothesize that in the context of the large APC/C assembly, the directly fused mAID degron cannot be recognized by SCF^{TIR1}. By binding to Venus, the mAID-nanobody might bridge the interaction between SCF^{TIR1} and mAID-Venus-ANAPC4. Hence, nanobody technologies such as the mAID-nanobody or deGradFP might be beneficial in such situations and therefore provide valuable alternatives.

- 3) We respectfully do not think however, that *“It would have been far more appropriate to compare the degradation of Venus-tagged ANAPC4 via the mAID-nanobody to degradation of directly mAID-tagged ANAPC4”*.

We show already in the experiments presented in Fig. 2 a comprehensive comparison between the classical auxin system and the mAID-nanobody. As shown in the manuscript, a key advantage of the mAID-nanobody is the ability to monitor the degradation of the targeted protein based on the fluorescent tag. This allows to i) identify the targeted cell by live cell microscopy and ii), directly relate the degree of depletion, as determined by the fluorescence readout, to the observed phenotype. Both features are not possible with direct AID-tagging and hence mAID-tagged ANAPC4.

For the deGradFP system the authors observed only a very low degradation of Venus-ANAPC4. Although in the text it is stated that an endogenous Venus-ANAPC4 was used, I suppose the authors used their 3xFlag-SBP-Venus tagged version of ANAPC4. Thus, it is conceivable that this construct is not very well recognized by the deGradFP nanobody construct. Further experiments to analyze this possibility (e.g. Co-IPs) would be needed.

- 1) We thank the reviewer for the suggestion to have a closer look at deGradFP degradation. Indeed, deGradFP was used to target 3xFlag-SBP-Venus tagged ANAPC4. To avoid naming complete tag throughout the manuscript, we have defined on p.9 “3xFlag-SBP-Venus-ANAPC4” as Venus-ANAPC4. Of note, 3xFlag-SBP-Venus-ANAPC4 represents endogenous ANAPC4 as all ANAPC4 alleles have been tagged.
- 2) Further, we noticed that deGradFP (which also contains the vhhGFP4 nanobody as the first-generation mAID-nanobody without lysines, see Supplementary Fig.1) is targeted for proteasomal degradation (Supplementary Fig. 6c, d). Indeed, without the proteasome inhibitor MG132, the expression levels of deGradFP are very low and hence might not be sufficient to remove all Venus-ANAPC4. Hence, it is conceivable that eventually an equilibrium between Venus-ANAPC4 and deGradFP expression and degradation is reached as also suggested by our quantification of Venus-ANAPC4 fluorescence in Fig. 5b. Transient expression of deGradFP as used in other studies (i.e. Caussinus et al.,) might alleviate this issue because in this case likely higher expression levels can be achieved. These findings re-iterate that dependent on the experimental system and the question at hand different degradation approaches have their advantages and disadvantages as now further highlighted in our discussion. We discuss these findings now on p. 19/20:

“We observed that deGradFP is strongly stabilized by proteasome inhibition (Supplementary Fig. 6c, d). Hence, similar to the first version of mAID-vhhGFP4 with lysine residues (Supplementary Fig. 1d, e), the vhhGFP4 containing deGradFP might be ubiquitinated and degraded upon interaction with the SCF. At least with the tetracycline-inducible system we used here, the achieved degree of deGradFP expression might not be sufficient to completely deplete Venus-ANAPC4.”

- 3) Degradation by deGradFP appears to be more heterogeneous in between cells. We now highlight this observation in the 24h time point in Fig. 5c by marking cells with efficient Venus-ANAPC4 degradation and we include a movie showing degradation by the AID-nanobody, the classical auxin system, and deGradFP in parallel (Supplementary Movie 2). This additional movie clarifies that deGradFP degrades Venus-ANAPC4 to some extent. We suspect that the heterogeneity in degradation we observed reflects the heterogeneity in deGradFP expression in between cells.
- 4) A small typo in the manuscript (p.14) that stated that Venus-ANAPC4 is degraded by ~25% has now been corrected as the correct value of degradation by deGradFP by the end of the experiment is ~35 % (Fig. 5).
- 5) As suggested by the reviewer we have used IgA beads to pull down the nanobodies from cells after induction with tetracycline (in the presence of MG132 to prevent degradation of deGradFP and Venus-ANAPC4). We observed that all nanobodies (mAID-nanobody, deGradFP^{ΔFbox} and deGradFP) efficiently pulled down Venus-ANAPC4, which means that the Venus epitope on Venus-ANAPC4 is recognized by deGradFP (see Figure to the reviewer). We hypothesized previously that the larger deGradFP might have difficulties to access its epitope on Venus-ANAPC4 in context of the APC/C. However, the IgA PD experiment shows that this is not the case and hence we have removed this interpretation.

Furthermore experiments (e.g. WB) which show the expression of the deGradFP are missing.

We now provide this data in Supplementary Fig. 6c, d. As mentioned above the expression of deGradFP is induced upon the addition of tetracycline. However, the overall levels of deGradFP compared to the mAID-nanobody and deGradFP^{ΔFbox} remain very low due to the fact that in deGradFP is degraded without proteasome inhibition.

Based on the data as presented I think the comparison between the three systems is not conclusive. To me a more thorough comparison of the mAID-nanobody approach with the two established techniques (AID, deGradFP) is mandatory to justify the claimed broad applicability and novelty - and consequently - publication of this additional application of the GFP-nanobody in Nature Communications.

We now addressed these points (see above). Altogether, we provide evidence that the AID-nanobody can target a multitude of proteins in different compartments and cellular structures. We present data on the degradation of a total of 10 proteins in human cell culture and zebrafish. Finally, we present a case in which the mAID-nanobody is clearly the method of choice by allowing the conditional and reversible inactivation of the APC/C that can be visualized in single cells by live cell microscopy.

Minor comments (but important)

What is the effect of lysine removal on the intracellular binding properties of the GFP-nanobody? The exchange of three lysine residues within a nanobody (115 aa) is quite a substantial modification. Thus a potential effect on binding should be carefully evaluated (comparing to wt nanobody), which become even more important considering the low effect of the deGradFP on Venus-ANAPC4, which might be due to a lower accessibility of the epitope.

As mentioned above and as seen in the Figure to the reviewer deGradFP binds well to Venus-ANAPC4 as judged by the IgA pulldowns. Notably, deGradFP^{ΔFbox} and deGradFP contain the same nanobody (namely with vhhGFP4) as the mAID-nanobody before lysines mutagenesis. The amount of ANAPC4 co-precipitated with all nanobodies directly correlates with the inputs indicating that no major differences exist between nanobodies for the ability to bind to Venus-ANAPC4.

Such experiments are also important for discussing the advantages of a trans-binding compared to cis-modified constructs, since a high affinity trans-binding nanobody might have dramatic effects on FP-labeled proteins regarding subcellular localization or dynamics (partially demonstrated by the authors themselves showing a cytoplasmic degradation of a GFP protein comprising a mitochondrial targeting sequence). A quite simple option to test intracellular binding would be intracellular IPs of cells co-expressing HA-tagged Venus and mAID-nanobody or mAID(K>R)-nanobody(K>R) using Protein A sepharose (the GFP-nanobody binds to protein A/G sepharose (Rothbauer et al., Mol. Cell. Proteomics 2008)).

- 1) We show in Supplementary Fig. 3b that persistent expression of the mAID-nanobody in Venus-ANAPC4 cells does result in not proliferation defects, does not change the localization of ANAPC4 (Supplementary Fig. 3c). In addition, we did not observe negative effects even when cells were cultured over several months.
- 2) The interpretation of the reviewer that the nanobody has an “dramatic effect” on mls-GFP “*partially demonstrated by the authors themselves showing a cytoplasmic degradation of a GFP protein comprising a mitochondrial targeting sequence*” probably results from a misunderstanding of the experimental setup: Cytoplasmic degradation is only induced upon the addition of auxin even though the nanobody is present in the targeted cells before.
- 3) As mentioned above we have already investigated the binding of the mAID-nanobody with lysine containing vhhGFP4 in the context of deGradFP^{ΔFbox} and deGradFP in the figure to the reviewers.

Figure 2 c and Figure 2 g)

WB data for mAID-nanobody should be included to visualize a potential effect of auxin treatment / removal on the nanobody.

This information was already provided in in the previous manuscript in particular in Supplementary Fig. 3e (at higher time resolution). See also Supplementary Fig. 3g and 3h.

Figure 3 a /b)

The recovery of Venus-LMNA and CCND1-Venus are very low. After removal of auxin, expression/fluorescence of both proteins only reaches 40 - 50% of their original level. The authors should comment on this.

Both constructs are expressed from the same (CMV) promotor, which appears to allow slower recovery in our experimental system than the endogenous INCENP promotor. We have added a comment on this result on page 8:

“Notably, CMV-driven Venus-LMNA and CCND1-Venus recovered slower than INCENP that was expressed under the control of endogenous promotor. This likely reflects differences in mRNA levels, translation efficiency and/or folding properties in between these proteins.”

Supplementary Figure 2 b)

How quantification of mAID-vhhGFP levels was performed? There is no explanation in the figure legends or in the Mat&Meths section.

We thank the reviewer for pointing out this oversight. mAID-vhhGFP levels were quantified by quantitative Western blotting using an near infrared detection-based LI-COR system. We have added this information to the method section and the figure legend.

Supplementary Figure 3 e/f

What was the basis for WB quantification? How many biological/technical replicates? There is no information in the figure legends / Mat&Meths section.

We thank the reviewer for pointing out this oversight. For quantification of Western blot data see point above see above. The data has been quantified from 3 independent experiments and is presented as the mean +/- s.e.m. as now indicated in the Figure legend.

Discussion (page 12)

What do the authors mean by “the small 11 kDa mAID-nanobody”? The fusion of the GFP-nanobody (13 kDa) to a HA-tag (1 kDa) and the mAID (7.5 kDa) results in a protein with a molecular weight of ~ 21.5 kDa (as shown by the authors in WB analysis (Suppl. Figure 1 e)). Since this statement is not correct, the advantages compared to the larger deGradFP (34 kDa) are somehow limited and have to be discussed more properly.

We thank the reviewer to point out this mistake. The predicted molecular weight of the mAID-nanobody is 26 kDa and it migrates in Western blot analysis at ~31 kDa. A previously mislabeled Western blot in Supplementary Fig.1 is now corrected.

Based on the result that the mAID-nanobody, deGradFP^{ΔFbox} and deGradFP all bind well to Venus-ANAPC4 (see Figure to the reviewers), we have removed this part of the discussion.

Mat&Meth

Do HeLa cells comprising 3 ANAPC4 alleles?

While we observed that both mAID-Venus-ANAPC4 clones we analyzed appear to have 3 alleles, we do not know if this is true for all HeLa cells. Of note, the Gerlich laboratory reported that Ki67 has 3 alleles in HeLa cells (doi: 10.1038/nature18610). Further, according to (doi: 10.1534/g3.113.005777) the most prevalent copy number state of genes is 3 in HeLa cells.

Reference #9

This ref. does not refer to the expression of nanobodies in living cells.

We thank the reviewer to point out his mistake, which is now corrected. The new reference #9 now refers to "Rothbauer, U. *et al.* Targeting and tracing antigens in live cells with fluorescent nanobodies. *Nat. Methods* **3**, 887–889 (2006)."

References #16 - #18

None of those publications refer to GFP-stock collections for cell culture models. E.g. Ratz et al, show mostly CRISPR-mediated heterozygous tagging of various proteins with GFP.

The reviewer is correct. These publications do not refer to stock collections and we have amended the wording to correct this mistake.

However, contrary to the statement of the reviewer the majority CRISPR-mediated taggings by Ratz et al. are homozygous (4 homozygous, 1 heterozygous). Lackner et al., performed tagging in haploid human HAP1 cells. Hence, here tagging of the only allele is sufficient. We have added further references showing further homozygous taggings in human cells: Of CCND1 in MCF10A cells by Goodkin et al. and of Ki67 in HeLa cells by Cuyen et al.

Figure for the reviewer:

mAID and deGradFP nanobodies co-precipitate Venus-ANAPC4. deGradFP and deGradFP^{ΔFbox} expression was induced with 1μg/ml tetracycline for a total of 10 hours. After 2 hours 10 μM of the proteasome inhibitor MG132 was added to prevent degradation of deGradFP and Venus-ANAPC4. Afterwards cells were harvested, extracts prepared, nanobodies precipitated with IgA beads followed by separation by SDS-PAGE, and Western detection with the indicated antibodies.

We consistently noticed that the mAID-nanobody was not stable at 4C in a prepared extract during the pull down resulting in α-VHH reactive cleavage product at 14 kDa (indicated by an arrow). This is consistent with a nanobody-only truncation of a predicted molecular weight of 13 kDa. The asterisk indicates a signal derived from the protein A beads. No such cleavage product is observed before the pull down or when cells are directly lysed in sample buffer (see Supplementary Figure 6), indicating that the mAID-nanobody is not cleaved in living cells.

All nanobodies, mAID-nanobody, deGradFP^{ΔFbox} and deGradFP (two independent clones), co-precipitate ANAPC4. No Venus-ANAPC4 is co-precipitated from the parent Venus-ANAPC4 cell line expressing no nanobodies.

REVIEWERS' COMMENTS:

Reviewer #1 (Remarks to the Author):

The authors have done careful revision of the manuscript and showed convincing data to clear my concerns. I feel that the current manuscript is adequate to be considered for publication.

Reviewer #3 (Remarks to the Author):

Review: „Conditional control of fluorescent fusion protein degradation by an auxin dependent nanobody“ Daniel et al.

While the authors could not detect a depletion of mAID-Venus-ANAPC4 they now claimed that the mAID-Venus-ANAPC4 is addressable by auxin dependent degradation in two mAID-nanobody knockin cell lines (new Suppl. Figure 5). This is only of moderate informative value. As it has been shown that the deGradFP system and the AID system are functional for depletion of various GFP- or AID-tagged proteins a meaningful comparison of the three systems is still lacking. In this context I do not understand what it means “the deGradFP (which also contains the vhhGFP4 nanobody as the first generation mAID-nanobody without lysines)...Does the deGradFP is not the original construct as described by Caussinus et al but comprises the nanobody without lysine residues?

From the WB analysis of the immunoprecipitation of the mAID-nanobody it becomes obvious that the interaction with its antigen and the stability of the mAID-nanobody is rather low compared the deGradFP. As mentioned previously this could be due to the exchange of the lysine residues. I wondered why the authors did not followed my previous suggestion and performed IPs using the mAID-nanobody or mAID(K>R)-nanobody(K>R)

Point-by-point response to the reviewers

We would like to thank the reviewers for their comments, which are addressed in the point-by-point response below.

Reviewer #1 (Remarks to the Author):

The authors have done careful revision of the manuscript and showed convincing data to clear my concerns. I feel that the current manuscript is adequate to be considered for publication.

We thank the reviewer for the positive evaluation of our manuscript.

Reviewer #3 (Remarks to the Author):

Review: „Conditional control of fluorescent fusion protein degradation by an auxin dependent nanobody” Daniel et al.

While the authors could not detect a depletion of mAID-Venus-ANAPC4 they now claimed that the mAID-Venus-ANAPC4 is addressable by auxin dependent degradation in two mAID-nanobody knockin cell lines (new Suppl. Figure 5). This is only of moderate informative value.

As requested by this reviewer and reviewer 1, we have presented in our previous revisions further evidence that mAID-Venus-ANAPC4 cannot be targeted by the classical auxin system, whereas simple transfection of the mAID-nanobody targets mAID-Venus-ANAPC4 efficiently for degradation (Supplementary Figure 5). This experiment proves that the observed inability of direct *mAID-Venus-ANAPC4* degradation is not a consequence of a general impairment of the auxin-induced degradation pathway in the cell lines and thus highlights a case, where the mAID-nanobody but not a direct fusion is preferable. Hence, as argued in the discussion dependent on the protein and/or the experimental system, each degradation system has its advantages and disadvantages and thus alternative approaches are highly beneficial.

As it has been shown that the deGradFP system and the AID system are functional for depletion of various GFP- or AID-tagged proteins a meaningful comparison of the three systems is still lacking.

We politely do not agree with this assessment. The key point of our manuscript is to present and carefully characterize a new technology and indicate future applications that cannot be or cannot be as proficiently addressed with existing technologies, such as deGradFP or the classical auxin system. In our manuscript, we have performed careful comparisons of the mAID-nanobody with the classical auxin system using GFP as an experimental model substrate in Figure 2 and between the mAID-nanobody, the classical auxin system and deGradFP in Figures 5, Supplementary Figure 5 and Supplementary Figure 6.

In this context I do not understand what it means “the deGradFP (which also contains the vhhGFP4 nanobody as the first generation mAID-nanobody without lysines)....Does the deGradFP is not the original construct as described by Caussinus et al but comprises the nanobody without lysine residues?”

We are sorry to have caused a confusion. The sentence should have been: “which also contains the vhhGFP4 nanobody as the first-generation mAID-nanobody **with** lysines”. As indicated in the methods section, the deGradFP construct we used was subcloned from the plasmid deposited by Caussinus et al., on Addgene and hence, the expressed GFP-nanobody is identical to their deGradFP. The anti-GFP nanobody used by Caussinus et al. (first published by Saerens, D. et al., 2005) is the basis for the lysine-less mAID-nanobody we developed. Thus, the nanobody

(vhhGFP4) in deGradFP is identical to the first generation of the mAID-nanobody containing all lysine residues (Supplementary Figure 1d.).

From the WB analysis of the immunoprecipitation of the mAID-nanobody it becomes obvious that the interaction with its antigen and the stability of the mAID-nanobody is rather low compared the deGradFP. As mentioned previously this could be due to the exchange of the lysine residues. I wondered why the authors did not followed my previous suggestion and performed IPs using the mAID-nanobody or mAID(K>R)-nanobody(K>R)

We politely do not agree with this interpretation. Compared to deGradFP the mAID-nanobody is apparently more stable than deGradFP in living cells. At steady state, deGradFP requires the addition of MG132 to prevent its proteasomal degradation, whereas the lysine-less mAID-nanobody is stable (see Supplementary Figure 6c, d). Further, the stabilization of the mAID-nanobody in the presence of auxin is a direct consequence of lysine mutagenesis (Supplementary Figure 1e).

The difference in the total levels of mAID-nanobody and deGradFP proteins after MG132 addition observed in Supplementary Figure 6 most likely reflects the fact that the expression deGradFP has been induced by 1 µg/ml tetracycline, whereas mAID-expression is only driven by the residual tetracycline present in the medium (see methods) - and hence is much lower (see also Supplementary Figure 2b). We choose this condition on purpose to exactly recapitulate the conditions used for the comparisons in Figure 5.

In our last revision, we already had performed an IP experiment comparing lysine-less vhhGFP4 to WT vhhGFP4 as suggested by the reviewer. In the figure to the reviewers, we have compared the ability of the lysine-less mAID-nanobody and deGradFP to co-precipitate the GFP antigen using Venus-ANAPC4 as a model. deGradFP contains the identical GFP-nanobody (vhhGFP4) (but with lysines) as the lysine-less mAID-nanobody and thus recognizes the same epitope. The IgA pull downs clearly show that despite less mAID-nanobody being pulled down by the IgA beads (likely reflecting the lesser amount of mAID-nanobody in the input and/or a lower affinity of the lysine-less vhhGFP4 to the IgA beads), a similar amount of Venus-ANAPC4 is co-precipitated compared to deGradFP pull downs. Thus, taken into account the ratio of IgA-bound nanobody and co-precipitated Venus-ANAPC4, the mAID-nanobody likely has a higher and not a lower affinity to the GFP epitope as suggested by the reviewer.

The observation that during the IP experiment the lysine-less nanobody appears to be proteolytically cleaved (Figure to the reviewers, last revision) *in vitro* does not concern the proposed application of the mAID-nanobody in living cells: In living cells, this cleavage does not occur as seen in the input before the IgA pull down (Figure to the reviewers, last revision) and also in Supplementary Figure 6c, where cells have been directly lysed in SDS-sample buffer.